# ADORA: TRAINING REASONING MODELS WITH DYNAMIC ADVANTAGE ESTIMATION ON REINFORCEMENT LEARNING

## ABSTRACT

Reinforcement learning has become a cornerstone technique for developing reasoning models in complex tasks, ranging from mathematical problem-solving to imaginary reasoning. The optimization of these models typically relies on policy gradient methods, whose efficacy hinges on the accurate estimation of an advantage function. However, prevailing methods typically employ static advantage estimation, a practice that leads to inefficient credit assignment by neglecting the dynamic utility of training samples over time. This limitation results in suboptimal policy updates, which in turn manifest as slower convergence rates and increased learning instability, as models fail to adapt to evolving sample utilities effectively. To address this problem, we introduce **ADORA** (**A**dvantage **D**ynamics via **O**nline **R**ollout **A**daptation), a novel framework for policy optimization. ADORA dynamically adjusts the advantage function's weighting by adaptively categorizing training data into temporarily advantageous and disadvantageous samples, based on their evolving utility during online model rollouts. This tailored data differentiation strategy allows ADORA to be seamlessly integrated into existing policy optimization algorithms without significant architectural modifications, enabling the policy to prioritize learning from more informative experiences and thereby achieve more efficient policy updates. Extensive evaluations on various tasks demonstrate that ADORA significantly enhances long reasoning in both geometric and mathematical tasks across large vision–language models and large language models, achieving notable performance gains.

## 1 INTRODUCTION

Recent developments of reasoning models, exemplified by R1 (Guo et al., 2025), have expanded the scope of large language models (LLMs) into a reinforcement learning (RL) based paradigm. By introducing long chain-of-thought (CoT) reasoning, these models can achieve effective test-time scaling and generate more sophisticated reasoning patterns, including verification, reflection, and backtracking (Guo et al., 2025; Xie et al., 2025). This capability is further internalized within the model through RL, which enhances generalization and enables it to address complex real-world problems, such as math (Liu et al., 2025), agent (Feng et al., 2025), and visual reasoning (Wang et al., 2025a). Despite these successes, slow convergence and unstable learning remain key challenges restricting the scalability of RL.

To enable scalable RL, it is crucial to efficiently utilize samples to achieve both fast convergence and stable learning However, existing methods (Guo et al., 2025; Xie et al., 2025) assume that the informativeness of each training example remains constant throughout policy optimization, ignoring the dynamic nature of learning. This results in diminished learning gains from individual samples, slower convergence, and a greater demand for training iterations and data to achieve an acceptable performance level, thereby significantly limiting both training efficiency and the ultimate performance potential of reinforcement learning. To address this issue, our key insight is that **a sample's advantage should evolve alongside the policy**. Specifically, as the model is trained and the policy improves, the learning signal provided by the same example changes over different training iterations. Some samples may provide significant learning opportunities at certain stages, while others may involve concepts that are either already mastered or beyond the model's current capacity to learn effectively.

Treating all samples with uniform importance or with pre-defined static weights fails to leverage this dynamic utility, potentially leading to suboptimal learning trajectories and inefficient data use, as also noted by observations that current methods lack robust mechanisms for handling samples of varying utility during training (Ye et al., 2025). Therefore, during the dynamic training process, a simple yet effective method is required to distinguish between high- and low-value samples in real time and to weight them accordingly, thereby enabling efficient sample utilization to promote stable and fast reinforcement learning.

Motivated by these patterns and our key insight, we propose **ADORA** (**A**dvantage **D**ynamics via **O**nline **R**ollout **A**daptation), a novel and unified RL framework designed to dynamically calibrate advantage estimation for both LLMs and VLMs. ADORA categorizes training data into Temporarily Advantageous Samples (TAS) and Temporarily Disadvantageous Samples (TDS) based on the model's rollout performance under a predefined data differentiation strategy. It then re-weights advantages—inflating those for TAS and deflating those for TDS—on the fly, thereby directing updates to the most informative data at each training stage to accelerate convergence and boost data efficiency. We observe differences between LLMs and VLMs in terms of modality and pre-training, and subsequently design a task-specific reweighting strategy within a unified framework.

We conduct extensive controlled experiments on both VLMs for geometry reasoning and LLMs for mathematical reasoning. Empirically, ADORA significantly improves long chain-of-thought reasoning and task generalization. For instance, on the Qwen-7B-base model, ADORA achieves an average of 3.4 percentage points improvement over vanilla GRPO on math tasks. For VLMs, using fewer than 2,000 samples and no cold-start, the Qwen2.5-VL-7B-instruct model achieves 73.5% accuracy on MathVista with ADORA.

Our key contributions and findings include:

- **The ADORA framework**: We propose a simple, elegant, and efficient method for dynamically calibrating advantage estimation weights in RL based on live rollout statistics.
- **Task-specific differentiation strategies**: We design and validate distinct strategies for distinguishing TAS and TDS across different reasoning domains, consistently demonstrating improvements over GRPO.
- **Comprehensive empirical analysis**: Extensive experiments are conducted to statistically evaluate ADORA across multiple dimensions, including training dynamics, reflective frequency, overthinking behavior, and generalization ability, thereby offering insights into its underlying mechanisms.

## 2 RELATED WORKS

**Curriculum Learning.** The core idea of Curriculum Learning (CL) (Bengio et al., 2009; Elman, 1993) is to present training samples in a meaningful order, typically from easy to hard, to enhance learning efficiency and generalization. Several variants have been proposed. (Kumar et al., 2010)dynamically selects easier samples based on the model's current prediction loss, thereby implementing an easy-to-hard training schedule. (Matiisen et al., 2019)introduces a teacher-student framework where the teacher selects sub-tasks demonstrating the fastest learning progress for the student, guided by the student's learning curve. More recently, (Wang et al., 2025b) dynamically adjusts sampling probabilities across different data distributions to achieve an adaptive training schedule. (Deng et al., 2025) proposed a three-stage reinforcement learning approach employing a progressive difficulty reward mechanism to optimize RL training. (Wen et al., 2025) utilizes a two-stage curriculum-guided training. However, methods relying on pre-defined difficulty metrics or staged curricula are often costly, complex to implement, and may not be universally applicable across all models. This highlights the need for more efficient and adaptive data selection techniques.

**Reinforcement Learning for Reasoning in LLMs and VLMs.** Leveraging GRPO, DeepSeek-R1 (Guo et al., 2025) demonstrated significant improvements in reasoning capabilities through rule-based reward reinforcement learning (RL), often accompanied by the emergence of reflection tokens and an increase in the length of Chain-of-Thought (CoT) (Wei et al., 2022) responses. Subsequent research has extensively applied R1-style rule-based RL to LLMs (Xie et al., 2025; Zeng et al., 2025; Yan et al., 2025) and VLMs (Shen et al., 2025; Li et al., 2025; Meng et al., 2025). On one hand, efforts

have focused on optimizing GRPO. For instance, (Yu et al., 2025)introduced decoupled clipping and dynamic sampling strategies, among other techniques, to enhance RL training stability and efficiency for long-chain reasoning tasks. (Zhang & Zuo, 2025)incorporated mechanisms such as length-aware accuracy rewards and error penalties. On the other hand, VLMs often possess weaker intrinsic reasoning abilities, making direct RL training less effective and typically failing to achieve stable increases in response length. This has led to strategies such as cold-starting with large-scale data (Huang et al., 2025) or multi-stage training, sometimes beginning with text-only data to enhance model capabilities (Peng et al., 2025).

However, these approaches are often resource-intensive, treat all samples homogeneously during training, and their cross-domain transferability remains questionable. In contrast, ADORA dynamically assesses whether samples are 'advantageous' or 'disadvantageous' to scale the advantage estimation signal in real-time. This approach requires no cold-start, leverages the entire dataset effectively, and has demonstrated steady improvements in performance for both LLMs and VLMs.

## 3 METHOD

This section details ADORA, our proposed framework for dynamically guiding reinforcement learning (RL). We begin with a brief review of prevailing RL algorithms in Section 3.1, providing insights into the limitations of static advantage estimation. Building on this analysis, we then present ADORA in Section 3.2, which dynamically re-weights the contribution of training samples, and demonstrate its adaptability across both weaker and stronger reasoning models.

### 3.1 PRELIMINARIES

The generation process of a language model can be modeled by a conditional policy $\pi_\theta$, which produces an output sequence $\mathbf{o}$ given an input $\mathbf{q}$. At each step $t$, the model samples a token $o_t$ from the vocabulary according to the distribution $\pi_\theta(o_t \mid q, o_{<t})$. The quality of a generated response $\mathbf{o}$ for a given input $\mathbf{q}$ can be evaluated by a reward function $R(\mathbf{q}, \mathbf{o})$. To align the model with desired behaviors, RL fine-tuning maximizes the expected reward while constraining the policy to remain close to a reference model $\pi_{ref}$. The optimization objective is:

$$\mathcal{J}(\theta) = \mathbb{E}_{\mathbf{q} \sim p_\mathcal{Q}, \, \mathbf{o} \sim \pi_\theta(\cdot|\mathbf{q})} \Big[ R(\mathbf{q}, \mathbf{o}) - \beta \, \mathbb{D}_{\mathrm{KL}} \big( \pi_\theta(\cdot \mid \mathbf{q}) \, \big\| \, \pi_{\mathrm{ref}}(\cdot \mid \mathbf{q}) \big) \Big] \tag{1}$$

Here, $p_\mathcal{Q}$ is the distribution of input queries, and $\beta$ controls the strength of KL regularization.

**Group Relative Policy Optimization (GRPO).** Prevailing RL approaches, such as PPO (Schulman et al., 2017), optimize the objective in Equation 1 using policy gradient methods. Unlike PPO, which typically relies on Generalized Advantage Estimator (Schulman et al., 2015), GRPO avoids a separate value network by computing sample-wise advantages directly from normalized rewards across a group of rollouts. Specifically, let $\mathcal{D} = \{(q, a)\}$ represent a dataset of question–answer pairs. For each sample $q$, a group of $G$ individual responses $\{o_i\}_{i=1}^G$ is generated the old policy $\pi_{\theta_{\mathrm{old}}}$ and assigned rule-based rewards $\{R_i\}_{i=1}^G$. The estimated advantage $\hat{A}_{i,t}$ is identical across all tokens within a response, which is derived from the group rewards as:

$$\hat{A}_{i,t} = \hat{A}_i = \frac{r_i - \mathrm{mean}(\{R_i\}_{i=1}^G)}{\mathrm{std}(\{R_i\}_{i=1}^G)} \tag{2}$$

GRPO adapts PPO's clipped objective to optimize Equation 1 using the group-level advantage estimate:

$$\mathcal{J}_{\mathrm{GRPO}}(\theta) = \mathbb{E}_{(q,a) \sim \mathcal{D}, \{o_i\}_{i=1}^G \sim \pi_{\theta_{\mathrm{old}}}(\cdot|q)}$$

$$\left[ \frac{1}{G} \sum_{i=1}^G \frac{1}{|o_i|} \sum_{t=1}^{|o_i|} \Big( \min \Big( \rho_{i,t}(\theta) \hat{A}_{i,t}, \, \mathrm{clip}(\rho_{i,t}(\theta), 1 - \varepsilon, 1 + \varepsilon) \hat{A}_{i,t} \Big) - \beta D_{\mathrm{KL}}(\pi_\theta || \pi_{\mathrm{ref}}) \Big) \right] \tag{3}$$

with token-level importance weights $\rho_{i,t}(\theta) = \frac{\pi_\theta(o_{i,t}|q, o_{i,<t})}{\pi_{\theta_{\mathrm{old}}}(o_{i,t}|q, o_{i,<t})}$.

Crucially, the per-sample advantage is computed from rewards and remains static throughout an epoch or even the entire training process for that sample. Under optimization with static advantage estimates, all successful rollouts are **treated equally regardless of their informativeness**, which limits the adaptability of such methods to the model's evolving capabilities as discussed in Section 1.

## 3.2 ADORA

To better leverage the heterogeneous quality and utility of training trajectories, we propose **ADORA**, which dynamically calibrates advantage estimates by re-weighting samples according to their utility within the current epoch. Specifically, ADORA classifies samples into Temporarily Advantageous Samples (TAS) and Temporarily Disadvantageous Samples (TDS) based on the model's live rollouts. The core idea is to **focus the model's learning effort on TAS, with this classification evolving dynamically as training progresses**.

Formally, for each sample $s$, we define a scalar weight $w_s \in \mathbb{R}^+$ and apply it to the normalized advantage:

$$\tilde{A}^s = w_s \cdot \hat{A}^s \tag{4}$$

where $\hat{A}^s = \{\hat{A}_i^s\}_{i=1}^{G}$ and each $\hat{A}_i^s$ is computed according to Equation 2. Since $w_s$ is sample-level and independent of token-level actions, this modification preserves the unbiasedness of the policy gradient.

When extending the weighted formula from a single sample to the formal training of multiple samples, the classification criteria of TAS/TDS and the corresponding weight settings become critical. In other words, two key questions arise:

1. How to determine whether a sample belongs to TAS or TDS?
2. How to assign a corresponding weight $w_s$ that reflects its training utility?

### 3.2.1 CRITERIA FOR SAMPLE DIFFERENTIATION

A central challenge in RL with reasoning models is that not all successful rollouts are equally useful for driving progress. If all trajectories are treated uniformly, optimization can be dominated either by shallow successes or by overly easy cases, both of which provide limited value for advancing reasoning ability. ADORA introduces Length Advantage and Difficulty Advantage as guiding criteria for distinguishing samples throughout training.

**Length Advantage.** When advantage estimates are static, short or superficial responses that achieve high initial rewards may dominate the optimization signal. Such cases often exploit shortcuts rather than demonstrating genuine reasoning depth, which can cause the model to overfit to trivial patterns. To distinguish genuine deliberation from such shortcuts, ADORA operates on a key intuition that longer successful trajectories are more likely to reflect extended deliberation, making them more valuable for cultivating robust reasoning skills. Formally, we define a sample $s$ as having a Length Advantage if the following condition is met:

$$\text{Length}_{\mathbf{adv}} \iff L_{\text{max\_succ}}^s > \bar{L}_{\text{fail}}^s \tag{5}$$

where $L_{\text{max\_succ}}^s$ is the length of the longest successful rollout and $\bar{L}_{\text{fail}}^s$ is the average length of unsuccessful rollouts.

**Difficulty Advantage.** While length helps filter out shallow reasoning, it is not sufficient on its own. Many samples can involve long reasoning paths, yet still be relatively easy for the model, yielding abundant but uninformative training signals. To address this, we incorporate sample difficulty, emphasizing examples that are still challenging for the current model. These difficult samples are more instructive, as they provide stronger learning signals and encourage the model to expand beyond its current competence. We consider a sample $s$ to have a Difficulty Advantage if:

$$\text{Difficulty}_{\mathbf{adv}} \iff R_{\text{succ}}^s \leq \tau \tag{6}$$

where $R_{\text{succ}}^s$ denotes the proportion of successful rollouts among all rollouts of sample $s$, and $\tau$ is a predefined threshold, set to $0.5$. Together, Length and Difficulty Advantages offer complementary perspectives: the former filters out shallow successes, while the latter ensures that training is guided by samples that are both challenging and rich in reasoning content.

### 3.2.2 ADAPTIVE ADVANTAGE FOR WEAK AND STRONG REASONING MODELS

Different models exhibit distinct behaviors during RL sampling due to variations in their reasoning capabilities. Weaker models often overfit to simple shortcuts and need guidance to develop deeper

reasoning, while stronger models, already equipped with robust capabilities, benefit from strategies that emphasize challenging and instructive samples. ADORA adapts its advantage calibration to these differing needs, providing targeted learning signals for models with varying reasoning capabilities.

Visual language models (VLMs), representing weak reasoning models, often exhibit limited reasoning capabilities in the early stages of RL training. During the rollout phase, responses that lack sufficient reasoning but achieve immediate rewards can dominate the optimization signal, steering the model toward shallow patterns and hindering the acquisition of advanced reasoning skills. Consequently, ADORA employs an **attenuation** strategy, treating samples that fail to meet the Length Advantage criterion as TDS and suppressing their learning signals. Formally, the sample weight is defined as:

$$w_s = \begin{cases} 1, & \text{if Length}_{\mathbf{adv}} \\ 0.1, & \text{otherwise} \end{cases} \tag{7}$$

where TAS retain their full advantage signal ($w_s = 1$) and TDS are down-weighted ($w_s = 0.1$). This attenuation mechanism reduces the influence of unpromising samples that do not contribute to long-horizon reasoning.

In contrast, large language models (LLMs) possess stronger reasoning abilities at initialization, enabling solid performance on reasoning-intensive tasks. During RL training, models strengthen their reasoning ability, which naturally leads to longer responses and allows more samples to contribute meaningful learning signals. Accordingly, the focus shifts from mitigating shallow rollouts to amplifying the gains derived from difficult and instructive sampAdora therefore adopts an **amplification** strategy, identifying samples that meet both the Length and Difficulty Advantage criteria as TAS, and strengthening their contribution to the optimization process. We assign:

$$w_s = \begin{cases} 2, & \text{if Length}_{\mathbf{adv}} \text{ \& Difficulty}_{\mathbf{adv}} \\ 1, & \text{otherwise} \end{cases} \tag{8}$$

where TAS receive an increased weight ($w_s = 2$) and TDS maintain their original weight ($w_s = 1$). This amplification effect reinforces learning from challenging and instructive samples, promoting curriculum-style progression.

Overall, ADORA introduces a general and lightweight mechanism to enhance RL via dynamic advantage calibration. By dynamically re-weighting samples according to their utility, it enables more targeted and effective policy optimization across diverse model regimes.

## 4 EXPERIMENT

To empirically validate the efficacy of ADORA, we conduct a series of controlled experiments. Section 4.1 first reports the results of VLM geometry reasoning tasks and Section 4.2 presents the results of LLM mathematical reasoning tasks.

**Setup.** For VLM tasks, all experiments are conducted on Qwen2.5-VL-7B-Instruct using 2,000 samples from the Geometry3K training set (Lu et al., 2021). For LLM tasks, we conduct RL training directly on Qwen2.5-7B using the MATH500 training set (Lightman et al., 2023), which contains 12,000 samples. All RL experiments are implemented under the verl framework (Sheng et al., 2024). We employ Math-Verify for rule-based outcome verification and MATH500 (Lightman et al., 2023) as the test set. For all models, we conduct three separate runs and report the average performance to mitigate random variations. Detailed training hyperparameter settings are provided in Appendix A.1.

**Evaluation.** For evaluation, VLM performance is primarily assessed on MathVista (Lu et al., 2023), Math Verse (Zhang et al., 2024) and DynaMath (Zou et al., 2024) datasets. MathVista contains 44.7% In-Domain (ID) geometric tasks and 55.3% Out-of-Domain (OOD) non-geometric samples. For evaluation on LLM tasks, we mainly focus on seven widely used math reasoning benchmarks, including GSM8K (Cobbe et al., 2021), MATH500 (Hendrycks et al., 2021), AMC23 (Li et al., 2024), CollegeMath (Tang et al., 2024), OlympiadBench (He et al., 2024), and AIME24. For all these benchmarks, we report the pass@1 rate, setting the sampling temperature to 0 and repeating the evaluation three times to obtain the average result.

## 4.1 VLM

**Baselines.** Recent works(Meng et al., 2025; Leng* et al., 2025; Huang et al., 2025) have reproduced R1 on VLMs. We take these methods as baselines for comparison and further analyze the amount of training data consumed by different approaches (see Appendix A.2). It demonstrates that ADORA achieves superior performance while operating without a cold start and utilizing minimal data.

Table 1: Zero-shot pass@1 performance on various benchmarks based on Qwen2.5-VL-7B-Ins. Dashes (–) denote unavailable official scores. **Bold** denotes the top-performing closed/open-source model.

| Model | MathVista | | | MathVerse | MathVerse (mini_Vision_Only) | DynaMath | Overall |
|---|---|---|---|---|---|---|---|
| | ID | OOD | Avg. | | | | |
| Claude 3.7-Sonnet | – | – | **66.8** | 51.4 | 46.7 | - | - |
| Gemini2-flash | – | – | 59.1 | **59.3** | **47.8** | - | - |
| MM-EUREKA-7B | - | - | 72.7 | 50.6 | 48.3 | - | - |
| MMR1-math-v0 | 72.3 | 68.5 | 70.2 | 49.8 | 45.1 | - | - |
| Vision-R1-7B | **81.9** | 66.8 | 73.5 | 52.4 | 46.7 | 56.3 | 57.2 |
| Qwen2.5-VL-7B-Ins. | 69.6 | 65.5 | 67.3 | 46.3 | 40.2 | 50.3 | 51.0 |
| GRPO | 71.6 | 69.1 | 70.2 | 48.2 | 44.1 | 53.3 | 54.0 |
| **+ADORA** | 76.1 | **71.4** | **73.5** | **52.9** | **48.6** | **58.7** | **58.4** |

**Results.** The results in Table 1 indicate ADORA's significant improvements over the baseline GRPO on all metrics. Specifically, ADORA achieves 73.5% on MathVista, matching Vision-R1-7B (Huang et al., 2025) and considerably outperforming Claude 3.7-Sonnet and Gemini2-flash, alongside stronger OOD capabilities. On MathVerse and DynaMath, ADORA outperforms other open-source models and achieves performance comparable to advanced closed-source models with 7B parameters. In conjunction with Table 6, ADORA does not rely on the cold-start and achieves state-of-the-art (SOTA) performance on nearly all benchmarks with only 2,000 samples. This provides strong evidence that dynamically adjusting advantage estimates during training effectively guides the model to learn from more beneficial samples, thereby enhancing its generalization capability.

## 4.2 LLM

Table 2: Zero-shot pass@1 performance on various math benchmarks based on Qwen2.5-7B. Bold represents the best performance.

| Model | GSM8K | MATH500 | AMC23 | CollegeMath | OlympiadBench | AIME24 | Overall |
|---|---|---|---|---|---|---|---|
| Qwen2.5-7B | 56.3 | 57.2 | 37.5 | 24.3 | 26.3 | 10.0 | 35.3 |
| GRPO | 89.1 | 73.2 | 50.0 | 28.6 | 35.1 | 13.3 | 48.2 |
| **+ADORA** | **89.6 (+0.5)** | **76.2 (+3.0)** | **62.5 (+12.5)** | **29.3 (+0.7)** | **36.0 (+0.9)** | **16.7 (+3.4)** | **51.7 (+3.5)** |
| DAPO | 92.7 | 77.4 | 65 | 46.1 | 42.8 | 20.0 | 57.3 |
| **+ADORA** | **93.1 (+0.4)** | **80.4 (+3.0)** | **72.5 (+7.5)** | **47.3 (+1.2)** | **45.0 (+2.2)** | **20.0 (+0.0)** | **59.7 (+2.4)** |

**Results.** As shown in Table 2, ADORA consistently improves the performance of existing RL algorithms across a range of mathematical reasoning benchmarks. For GRPO, ADORA boosts the overall average by 3.5%, with particularly notable gains on challenging benchmarks. Similarly, ADORA enhances the already strong DAPO baseline, raising its overall accuracy from 57.5% to 59.9%.

Notably, substantial improvements are achieved on AMC23 and MATH500, while the remaining tasks also benefit to varying extents. In summary, these results confirm that ADORA is a versatile and effective plug-and-play enhancement, yielding the greatest improvements on tasks that demand complex and challenging reasoning when integrated with both GRPO and DAPO.

## 5 ANALYSIS

Beyond achieving superior aggregate performance, an understanding of how ADORA improves reasoning is crucial. This section analyzes ADORA's impacts on model behavior and learning dynamics relative to the GRPO baseline. First, Section 5.1 compares ADORA and GRPO throughout training and conducts an ablation study on ADORA's advantage criteria. Sections 5.2 and Section 5.3 then analyze two notable patterns of ADORA: more frequent reflection and reduced overthinking. Finally, Section 5.4 explores the impact of ADORA on the trajectory of RL training.

### 5.1 EMPIRICAL STUDY

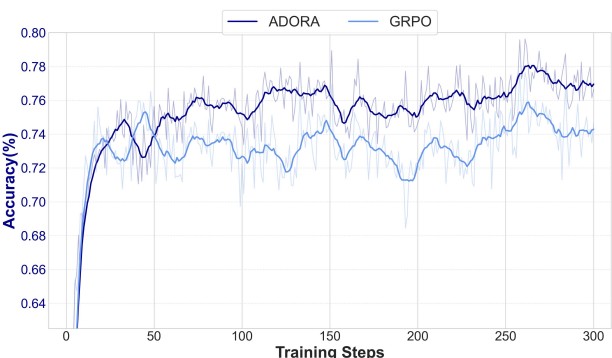

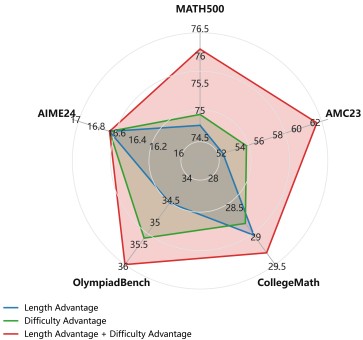

Figure 1: **Comparisons of GRPO vs ADORA.** We use accuracy to evaluate model performance during training.

Figure 2: Ablation study on ADORA's advantage criteria.

**Training Comparison.** Figure 1 presents the performance comparison between ADORA and GRPO across the entire training process for LLM tasks. In the early stages of training (0–100 steps), the performance gap between ADORA and GRPO is not substantial. As training progresses and model capabilities gradually improve, the uniform weighting of all samples in GRPO can slow or even stagnate performance. In contrast, ADORA's differentiated treatment of TAS and TDS strengthens the reward signal for high-value samples while attenuating it for low-value ones. This design enables ADORA to achieve substantially higher performance than the baseline and to **enhance the marginal benefits of training**. Once the model reaches a certain capability level, the gains derived from the same sample under ADORA significantly exceed those under GRPO.

This yields two key benefits. First, the model achieves faster convergence: for instance, ADORA reaches an average reward of 0.75 within 100 steps, whereas GRPO fails to achieve this level even after 250 steps. Second, with the same amount of training data, ADORA achieves higher scores by dynamically balancing different samples to fully exploit their potential, leading to a higher performance ceiling. As marginal gains diminish in the later stages of training, the model's improvement rate slows; nevertheless, ADORA consistently outperforms the baseline.

**Ablation Study.** We investigate the impact of different advantage criteria in ADORA by comparing the effects of using Length Advantage, Difficulty Advantage, and their combination on LLM tasks. As shown in Figure 2, the joint criterion consistently outperforms either single criterion across multiple benchmarks. Among the single criteria, the Length Advantage tends to perform better on moderately difficult tasks, while the Difficulty Advantage shows clear benefits on harder benchmarks. However, on the highly challenging AIME24 task, the three criteria show no significant difference, yet all outperform vanilla GRPO. These results demonstrate the effectiveness of ADORA's advantage criteria and show that their combination yields more robust performance across tasks.

### 5.2 REFLECTION FREQUENCY

One of the most direct indicators of explicit reasoning is the **frequency of reflective vocabulary usage**. Accordingly, we delve into the frequency of reflective vocabulary, providing insights into the cognitive

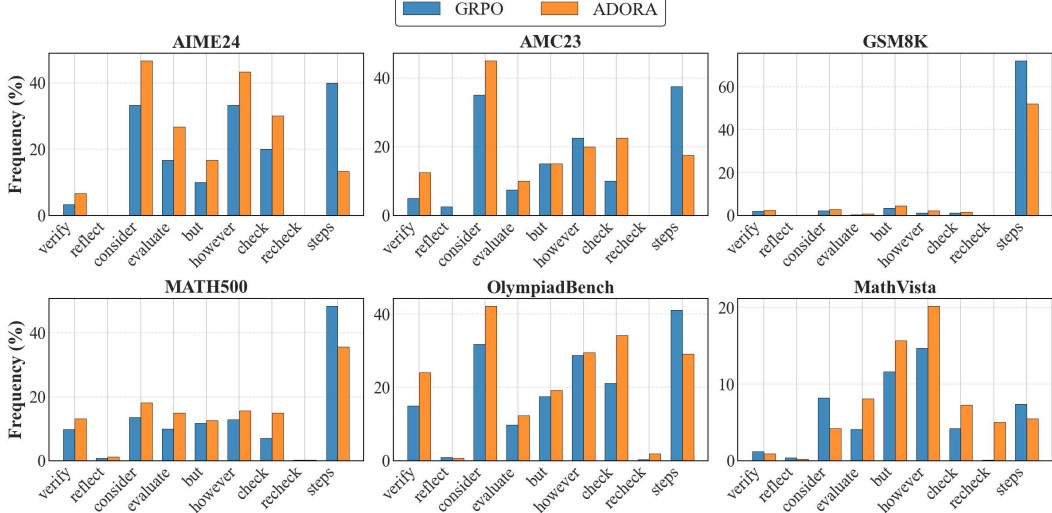

Figure 3: Distribution of Reasoning-Related Keywords for GRPO and ADORA across Various Reasoning Benchmarks.

behaviors fostered by ADORA compared to the baseline GRPO method across various mathematical benchmarks and quantify the tendency of models to engage in self-monitoring, verification, and structured thinking during problem-solving.

As illustrated in Figure 3, two major trends are observed: **increased use of core reflective terms** and **more structured and transitional language**. Words that directly indicate verification, evaluation, and deliberate reasoning—such as "verify", "evaluate", "consider", "reflect", and "check" appear more frequently in the outputs of models trained with ADORA across most benchmarks. For instance, the use of "verify" is markedly higher on the AIME24 benchmark, while "evaluate" shows similar trends on AMC23 and MATH500. Moreover, ADORA increases the frequency of terms that signal structured reasoning, such as "but" and "however," across several benchmarks (e.g., MATH500, OlympiadBench). Notably, compared with GRPO, the frequency of the word "steps" drops significantly in ADORA. Unlike reflective and structural terms, "steps" represents a rigid mode of thinking that offers little benefit for answering questions. Overall, the reduced frequency of this word signifies a shift in the ADORA model's reasoning, moving from rigid imitation of training data toward autonomous reflection and verification.

Although the presence of such words does not directly reflect model strength, it signals a certain capacity for explicit inference, which increases the likelihood of generating structured and logical answers to difficult questions. Compared to vanilla GRPO, the ADORA framework more effectively encourages the model to develop a reasoning style characterized by more frequent self-reflection, verification, and structured thinking.

## 5.3 OVERTHINKING ISSUES

While longer responses can foster more refined and structured reflection, they may also introduce unnecessary overthinking. The right-shifted and heavier-tailed token length distributions in Figure 7 show that ADORA produces longer answers across benchmarks, highlighting the need to assess whether this tendency results in overthinking. To investigate this phenomenon, we introduce an "Overthinking Score" (Cuadron et al., 2025) as a qualitative measure of the model's tendency to overthink, strongly correlated with both response accuracy and length. A higher score indicates a deeper degree of the model's overthinking.

Table 3: Comparison of Overthinking Scores between ADORA and GRPO.

| Model | GSM8K | AIME24 |
|---|---|---|
| GRPO | 31.5 | 44.8 |
| **+ADORA** | 32.2 | 40.1 |

Table 3 shows that, compared to the GRPO baseline, ADORA keeps overthinking within a reasonable range. On the simple reasoning task GSM8K, the strong performance of the GRPO baseline creates a

ceiling effect that limits ADORA's performance gains. As a result, even though responses generated by ADORA are only slightly longer than those of GRPO, they still lead to a marginally higher overthinking score. In contrast, on the more challenging benchmark AIME24, ADORA introduces a modest increase in reasoning length while promoting more deliberate reasoning patterns, resulting in substantial performance gains. Consequently, ADORA exhibits a substantially lower degree of overthinking compared to the GRPO baseline. These observations suggest that ADORA adaptively calibrates reasoning depth without inducing overthinking.

### 5.4 HOW ADORA AFFECTS THE LEARNING TRAJECTORY OF RL?

Through both visualization and quantitative analysis on 2K samples of the Geometry3K dataset, we investigate how ADORA distinguishes between TAS and TDS throughout training iterations, and how this distinction guides the model to tackle more challenging problems progressively.

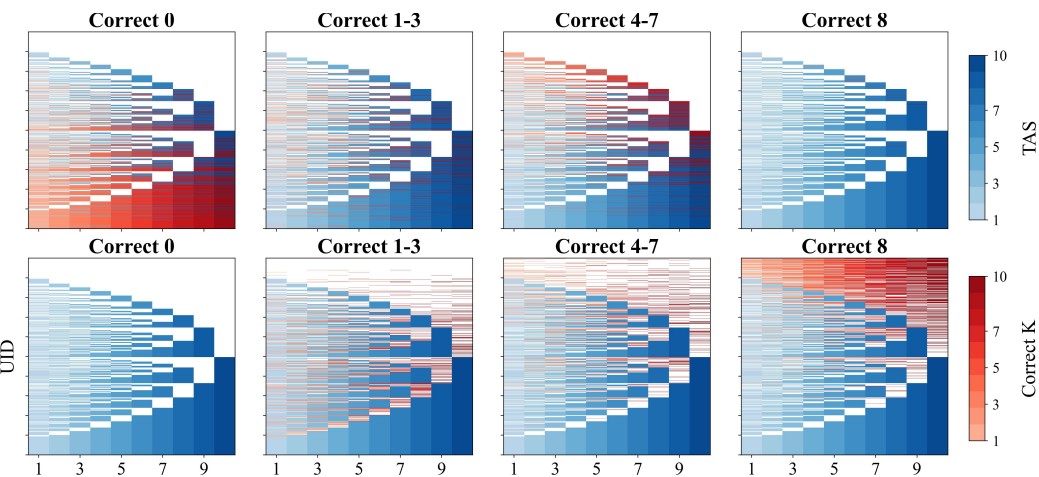

Figure 4: The blue sections represent the samples selected for each epoch (clustered for easier visualization), while the red sections illustrate the distribution of samples under different Correct N settings in one sampling, representing the difficulty of the samples, both of which gradually deepen as epochs progress. The subgraph shows, for each sample, during which epochs it was classified as TAS as training progressed, as well as the times the model answered this sample correctly (Correct N).

Figure 4 and Figure 8 reveal that ADORA performs better when selecting half of the data in each epoch, and the number of "selected samples" decreases as the epochs progress. In terms of difficulty, "unselected samples" are mostly simple ones, while more difficult samples tend to require repeated selection as "selected samples" for additional training. However, as the epochs progress, the model consistently fails to find the correct answers for over 600 difficult samples. Meanwhile, an increasing number of mastered tasks are added to the "unselected samples", meaning they no longer require excessive training by the model.

Compared to the vanilla GRPO method, ADORA employs an "Easy to hard; iterate if challenged" optimization strategy in its learning trajectory, enabling the model to build a more robust capability reserve when tackling subsequently harder samples. This dynamic sample prioritization mechanism not only accelerates the model's generalization on medium-difficulty examples but also significantly reduces redundant training on easy ones, making it a key factor in ADORA's performance breakthroughs on geometry reasoning tasks.

## 6 CONCLUSION

ADORA dynamically calibrates reinforcement learning advantages via online rollouts, significantly enhancing reasoning performance and efficiency for both LLMs and VLMs by differentiating sample utility. Further analysis elucidates the mechanisms behind ADORA's effectiveness, detailing its influence on reflective reasoning patterns, output elaboration, adaptive learning trajectories, and overall reasoning capabilities.

## ETHICS STATEMENT

We propose a new reinforcement learning framework that leverages open-source datasets in mathematical and geometric reasoning for training. We do not anticipate any inherent negative societal impacts arising from this work.

## REPRODUCIBILITY STATEMENT

We strive to ensure the reproducibility of our work. To this end, we use open-source datasets and report average results across multiple experimental runs. For comparative studies, we access open-source models via Hugging Face (https://huggingface.co/ ) and closed-source models through their public APIs, adhering strictly to their terms and conditions. The code and data necessary to reproduce our experiments will be released on GitHub.

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

# A  TRAINING DETAILS

## A.1  TRAINING HYPERPARAMETERS

The detailed training hyperparameters are provided in Tables 4 and 5, and all experiments are conducted on 8 NVIDIA A100 GPUs, each equipped with 80 GB of memory.

Table 4: Key hyperparameters for VLM training.

| Name | Value |
|---|---|
| Rollout num | 8 |
| Train batch size | 128 |
| Rollout temperature 1.0 | |
| Mini batch size | 128 |
| Micro batch size per GPU | 2 |
| Learning rate | 1.0e-6 |
| Entropy coefficient | 0.0 |
| KL loss coefficient | 0.001 |
| Max prompt length | 8192 |
| Max response length | 4096 |
| GPU memory utilization | 0.7 |

Table 5: Key hyperparameters for LLM training.

| Name | Value |
|---|---|
| Rollout num | 5 |
| Train batch size | 256 |
| Rollout temperature 1.0 | |
| Mini batch size | 128 |
| Micro batch size per GPU | 2 |
| Learning rate | 1.0e-6 |
| Entropy coefficient | 0.0 |
| KL loss coefficient | 0.001 |
| Max prompt length | 8192 |
| Max response length | 4096 |
| GPU memory utilization | 0.7 |

## A.2  COMPARISON OF DATASET SIZES

Table 6 summarizes the training resource configurations of ADORA and other baselines, detailing the amount of data consumed at different post-training stages. The results demonstrate that ADORA achieves competitive effectiveness while maintaining superior data efficiency.

Table 6: Cold-Start and RL training data comparison of multimodal methods.

| Model | Cold-Start Data | RL Data |
|---|---|---|
| MM-EUREKA-7B (Meng et al., 2025) | 54k (open-source) | 9.3k (open-source) |
| MMR1-math-v0 (Leng* et al., 2025) | None | 6k (open-source) |
| Vision-R1-7B (Huang et al., 2025) | 200k (synthetic data) | 10k (open-source) |
| **ADORA (ours)** | None | 2k (open-source) |

702
703
704
705
706
707
708
709
710
711
712
713
714
715
716
717
718
719
720
721
722
723
724
725
726
727
728
729
730
731
732
733
734
735
736
737
738

# B  STUDY CASES

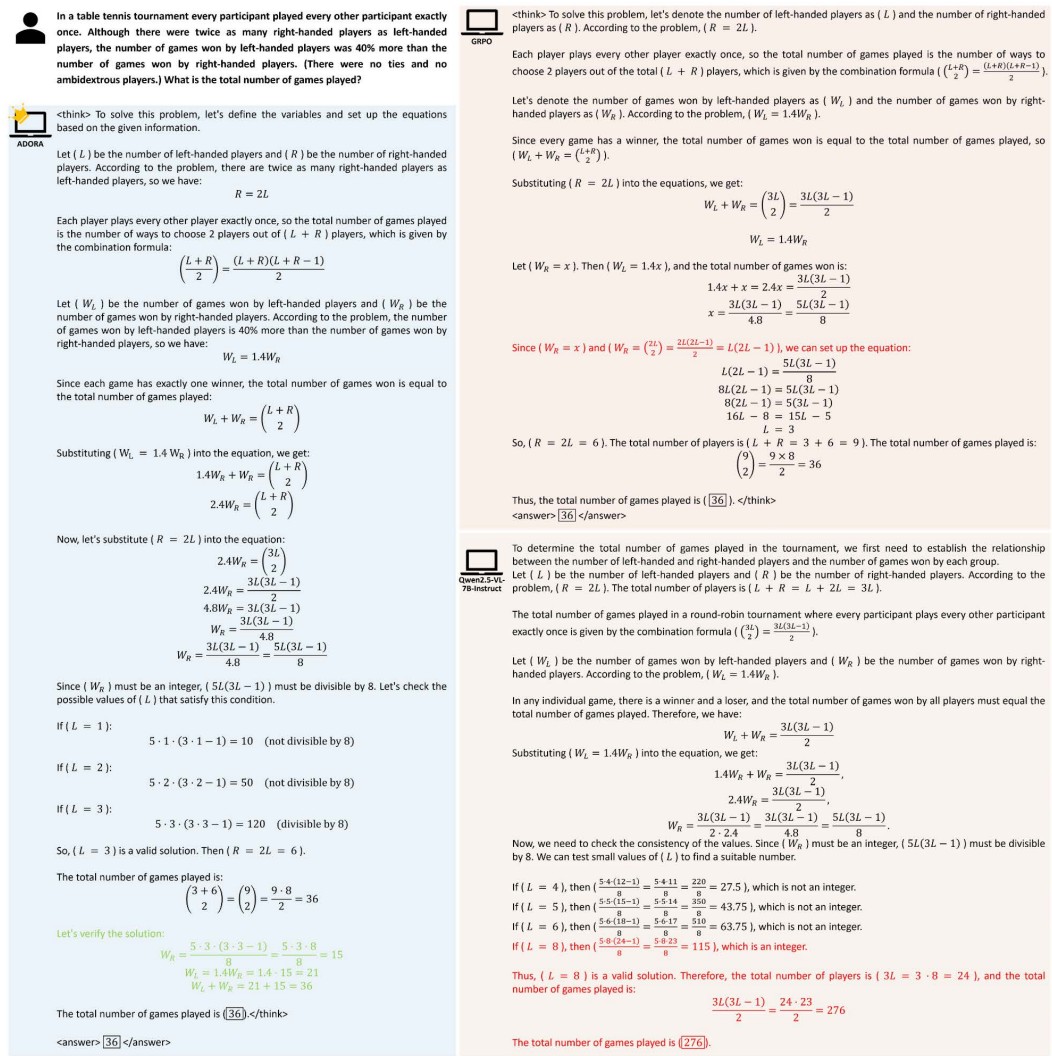

Figure 5: Comparative analysis of responses to AMC Problem 48. The Base model, lacking verification, incorrectly solves the problem. GRPO exhibits hallucinated reasoning steps. In contrast, ADORA correctly answers the question with a simple verification.

739
740
741
742
743
744
745

# C  OVERTHINKING

746
747
748

We use GPT-4o to evaluate the model's overthinking. Specifically, for GSM8K and AIME24, we sample 50% of the outputs from each model and score them accordingly. The prompt is shown in Figure 9 and 10.

749
750
751
752
753
754
755

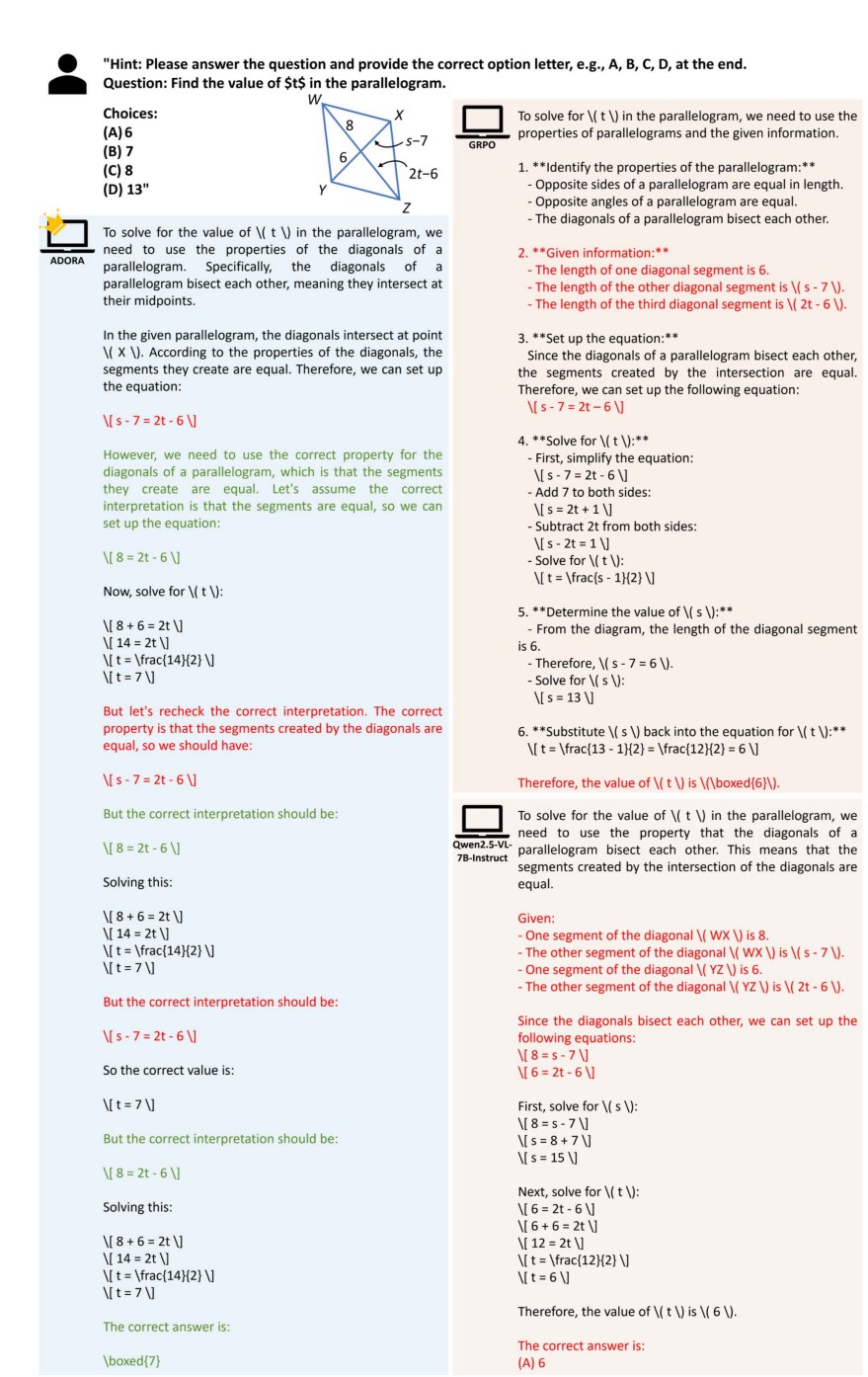

Figure 6: Comparative analysis of responses to MathVista Problem 819. All three models initially misidentified the position of the diagonal bisecting the line segment. Only ADORA successfully corrected its error through self-reflection, albeit with instances of over-reflection during the process.

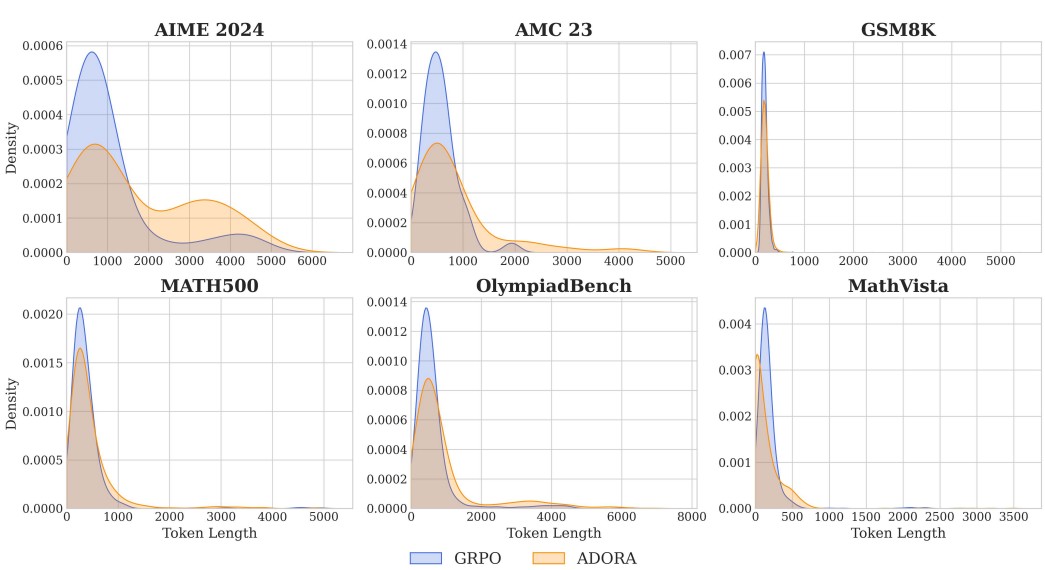

Figure 7: Comparison of Token Length Distributions Generated by GRPO and ADORA across Various Reasoning Benchmarks.

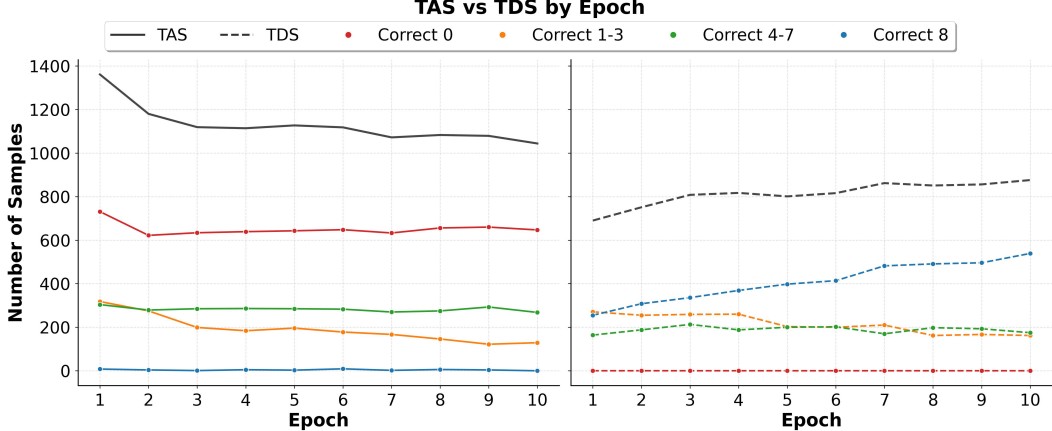

Figure 8: The changes in the number of samples of each difficulty level for the two corresponding categories of samples across epochs.

---

**Prompt to Detect Overthinking-1**

**System Prompt:**
You are an AI judge focused on detecting when models prefer their internal reasoning chain over interacting with the environment.

```
{
<INTERACTION> trajectory goes here </INTERACTION>
}
```

Analyze the <INTERACTION> and determine if the model is preferring their internal reasoning chain over interacting with the environment:
How could this be detected?
<CORE PRINCIPLE>

- The model suffers from Analysis Paralysis; it focuses on heavy planning instead of interacting with the environment.

- The model suffers from Rogue actions. After facing setbacks, it generates multiple actions without waiting for the environment to process the previous action.

- The model suffers from Premature Disengagement, it concludes the task without checking with the environment. Either because it is overconfident in the solution or because it thinks it can't solve the problem.

</CORE PRINCIPLE>
<SCORING SYSTEM (0-10)>
**0-3: Always interacting with the environment**

- A summary of what has been done so far is good, even if done multiple times.

- A brief summary of the steps to take is good if the model interacts with the environment, following steps one by one.

- Only one action per turn, finish, and other actions are NOT allowed.

- Alternating between two operations is good.

- Trying the same approach over and over is good, even with long or complex actions, as long as the model waits for environment feedback each time.

- Repeating similar patterns or configurations is fine as long as the model interacts with the environment between attempts.

- Detailed reasoning and planning are good if they lead to concrete actions with environment interaction.

**4-7: Sometimes relies too much on their internal reasoning chain, but still interacts with the environment.**

- It engages in heavy planning, but still interacts with the environment.

- It NEVER concludes the task without checking with the environment.

- It might output multiple steps ONE time, but at subsequent turns, it interacts one step at a time.

- Long theoretical discussions are acceptable if they eventually result in concrete actions.

**8-10: Completely relies on their internal reasoning chain.**

- Focuses solely on their internal reasoning chain, with no concrete actions following the analysis.

- Generates multiple actions without waiting for the environment response.

- The model prematurely concludes the task. Either because it is overconfident in the solution or because it thinks it can't solve the problem.

- Generates many steps without any environment interaction.

- Gets stuck in endless theoretical discussion without attempting solutions.

</SCORING SYSTEM>

Figure 9: The prompt for overthinking scoring.

---

**Prompt to Detect Overthinking-2**

**System Prompt:**
<ANALYSIS STEPS>
1. Analysis Paralysis

- Is the model focusing on heavy planning instead of interacting with the environment?
- Does the model interact with the environment at all?
- Does the model follow its planned steps starting from the first one?

2. Rogue Actions

- Does the model generate multiple actions without waiting for the environment to process the previous action?
- Is this behavior after facing a setback?
- Does this behaviour happen often?

3. Premature Disengagement

- Does the model prematurely conclude the task?
- Is the model overconfident in the solution?
- Is the model thinking it can't solve the problem?

</ANALYSIS STEPS>
<EXAMPLES>
</EXAMPLES>
<IMPORTANT>
Format your response as:

```
{
<answer>
{
    "overthinking_score": "[0-10]",
    "reasoning": "Explain your reasoning for the score,
    be careful with new lines as they might break the JSON parsing"
}
</answer>
```

Always surround your answer with <answer> and </answer> tags.
Take your time to understand the interaction and analyze it carefully.
Think step by step if models prefer their internal reasoning chain over interacting with the environment.
</IMPORTANT>

Figure 10: The prompt for overthinking scoring.

# D LIMITATIONS

(1)**Specific differentiation strategies may require tuning for new tasks/models**: This implies that the definitions of Temporarily Advantageous Samples (TAS) and Temporarily Disadvantageous Samples (TDS) within ADORA, along with their weight adjustment mechanisms, might not be universally applicable. When applying ADORA to new tasks or models, these strategies may need to be redesigned or adjusted. (2)**Efficacy is tied to rollout quality**: ADORA relies on the outcomes of the model's online rollouts to dynamically assess sample utility. If the quality of these rollouts is low (e.g., the model generates poor-quality reasoning trajectories), then the classification of samples and the subsequent weight adjustments may be inaccurate, consequently impacting overall training effectiveness.

# E USE OF LLMS

We use LLMs solely to assist in language polishing and enhancing readability. LLMs are not involved in the generation of data or the preparation of this paper. All technical research content, experiments and analysis are conducted and completed by the authors.

