# OpenReview forum: "ADORA: Training Reasoning Models with Dynamic Advantage Estimation on Reinforcement Learning"
_ICLR.cc/2026/Conference — ICLR 2026 Conference Withdrawn Submission_

### Official Review · Reviewer_nDXV · 2025-10-28

**Soundness:** 2
**Presentation:** 2
**Contribution:** 2
**Rating:** 4
**Confidence:** 5

**Summary:**

The paper proposes ADORA, a reinforcement learning framework that dynamically updates the advantage estimation weights during training.
The basic idea is to categorize samples into temporarily advantageous and temporarily disadvantageous based on their evolving utility, and ADORA enables more efficient policy optimization. Experiments across both large language models and vision–language models shows consistent gains over baselines like GRPO and DAPO, especially in reasoning-intensive tasks such as mathematical and geometric problem solving.

**Strengths:**

-  Feasible approach to dynamic weighting: The paper introduces a conceptually clean yet empirical effective way to dynamically calibrate the advantage function during reinforcement learning. It targets solving a key problem with static estimations.
-  Strong empirical validation: The experiments are relative extensive, covering both LLMs and VLMs. It shows clear and consistent performance improvements even with limited data. This gives confidence in the robustness of ADORA’s approach. However, in terms of ablation, there remain several concerns, which will be detailed next. (See Weakness)

**Weaknesses:**

- Generalization not deeply explored: Although ADORA performs well on tested benchmarks, the discussion on transferability and performance on out-of-distribution or unseen domains feels somewhat limited. Also, the weighting is based on heuristic rule and there lacks some theoretical insights.
- Dependence on rollout quality: The authors themselves note that ADORA’s success depends on the quality of generated rollouts (Appendix D). However, the paper does not clearly propose methods to mitigate low-quality samples or noise in the rollout process.
- Missing some prompt difficulty related works: There are related works that can be discussed in the manuscript. For example, MoPPS [1] actively infers the prompt difficulty and selects the subset to improve both efficiency and performance. SPO does a similar thing from policy optimization perspective.
- Besides the above, I have raised some questions below, which should be well addressed in either added discussions or experiments.

References:

[1] Qu, Yun, et al. "Can prompt difficulty be online predicted for accelerating rl finetuning of reasoning models?." arXiv preprint arXiv:2507.04632 (2025).

[2] Xu, Zhongwen, and Zihan Ding. "Single-stream policy optimization." arXiv preprint arXiv:2509.13232 (2025).

**Questions:**

- How sensitive is ADORA to the hyperparameters like τ and ws? Would small changes significantly affect convergence?
- How does ADORA behave with other family of base models such as Qwen and other sizes such as 1.5B base models?
- It seems reweighting mechanism adjusts the learning rate, so is the VLA result’s advantage from the learning rate adjustment.

Other suggestions:

- (1) It would be better to include an illustration Figure to show the detailed reweighting implementation steps in rollout and policy optimization.
- (2) Learning curves for all results should be reported if necessary.

---

> ### Author Response · Authors · 2025-12-02
>
> We thank Reviewer nDXV for acknowledging ADORA as a "feasible" and "empirically effective" approach. Your constructive comments regarding **generalization**, the **nature of reweighting**, and **related works** have helped us significantly strengthen the paper.
>
> > **W1: Generalization not deeply explored: Although ADORA performs well on tested benchmarks, the discussion on transferability and performance on out-of-distribution or unseen domains feels somewhat limited. Also, the weighting is based on heuristic rule and there lacks some theoretical insights.**
>
> Thank you for the reviewer’s comment. Our work does evaluate generalization beyond the training distribution, and the results show that ADORA improves transferability for both VLMs and LLMs.
>
> **On the VLM side**, the model is trained solely on **2k Geometry3K samples** containing only geometric diagrams, whereas MathVista includes 55.3% non-geometric OOD tasks with natural images, plots, and diverse visual layouts. Despite this clear train–test distribution gap, ADORA improves both the **ID and OOD splits** and raises Qwen2.5-VL-7B-Instruct to **73.5%** overall using far less data than prior methods, indicating that dynamic advantage calibration does not overfit to the geometric domain.
>
> **On the LLM side**, MATH500 focuses on algebraic and geometric competition problems, while our evaluation spans **six heterogeneous benchmarks—from GSM8K word problems to AMC, college-level exams, and OlympiadBench/AIME24—covering reasoning styles** not present in the training set. ADORA achieves its largest gains on the most distribution-shifted benchmarks, **such as AMC23 (+12.5) and AIME24 (+3.4)**, suggesting improved robustness rather than task-specific specialization.
>
> **Section 5** further shows that **ADORA promotes more deliberate and stable reasoning patterns**, which directly support cross-domain generalization. **While our evaluation already covers diverse OOD settings**, we agree that extending to additional domains such as science QA and multi-hop reasoning would further strengthen the conclusions, and we will include these in the revision.
>
> Regarding the **theoretical motivation** behind ADORA’s weighting scheme, our design follows established principles in policy gradient optimization.
>
> Short but reward-aligned trajectories often dominate early GRPO updates and **lead to low-information, low-variance gradients**. Length Advantage mitigates this by emphasizing trajectories that provide richer exploration of the underlying reward landscape. Difficulty Advantage acts as an adaptive form of importance weighting: gradients are most informative when the success probability is **neither too high nor too low**, where the policy’s output is most sensitive. Emphasizing this regime stabilizes the effective step size and keeps learning focused near high-curvature regions of the objective. Together, the two criteria produce an online curriculum that emerges naturally from the rollout distribution.
>
> This view aligns with our empirical findings that **ADORA accelerates convergence and yields smoother optimization dynamics**. We will make this rationale clearer in the revised manuscript.
>
> > **W2: Dependence on rollout quality: The authors themselves note that ADORA’s success depends on the quality of generated rollouts (Appendix D). However, the paper does not clearly propose methods to mitigate low-quality samples or noise in the rollout process.**
>
> The reviewer correctly identifies the dependence on rollout quality. We emphasize that addressing this dependence is the core motivation for ADORA's VLM strategy, specifically the Attenuation mechanism.
>
> **Denoising via Attenuation**: Rollouts from weaker models (e.g., VLMs) frequently contain low-quality or hallucinated reasoning steps. ADORA actively addresses this by identifying shallow reasoning paths (via the Length Criterion) and applying a strong attenuation weight **$w_s=0.1$**. This mechanism serves as an **active noise suppressor**, robustly preventing the policy from being misdirected by unreliable or low-utility trajectories. Our ablation studies conclusively demonstrate that this attenuation is a **critical component** for achieving stable and effective training with weaker base models.

---

> ### Author Response · Authors · 2025-12-02
>
> > **W3. It seems reweighting mechanism adjusts the learning rate, so is the VLA result’s advantage from the learning rate adjustment.**
>
> We appreciate the request for clarification on the mechanism. The distinction between ADORA and a global Learning Rate (LR) adjustment is fundamental:
>
> *   **Sample-Wise vs. Global**: A global LR scales the update **uniformly** across the entire batch. In contrast, **ADORA applies a sample-wise dynamic reweighting mechanism**, offering fine-grained control over the optimization process.
>
> *   **Gradient Direction**: ADORA does not merely adjust the step size; it fundamentally alters the **direction** of the gradient. By amplifying Temporarily Advantageous Samples (TAS) and attenuating Temporarily Disadvantageous Samples (TDS), ADORA ensures the policy update is guided primarily by **high-utility, instructive experiences**, shifting the effective gradient vector.
>
> *   **Stability and Efficiency**: While simply increasing the LR in methods like GRPO often leads to optimization instability, ADORA achieves **faster and more stable convergence (as shown in Figure 1)**. This is because it simultaneously applies an effective "high learning rate" to high-quality samples and a "low learning rate" to noisy samples within a single, stable optimization step.
>
> > **Q1. How sensitive is ADORA to the hyperparameters like τ and ws? Would small changes significantly affect convergence? ($\tau, w_s$)**
>
> We performed a comprehensive grid search (results in Appendix tables), showing that ADORA is highly robust:
>
> *   **Threshold $\tau$:** The 0.5 threshold for "Difficulty Advantage" provides a stable trade-off between easy (GSM8K) and hard (AIME24) tasks, effectively distinguishing "mastered" from "unmastered" samples. Ablation over **$\tau \in {0.25, 0.5, 0.75, 1}$** shows that all settings converge reasonably well, with $\tau = 0.5$ achieving the best overall balance:
> | $\tau$ | GSM8K  | MATH500  | AMC23 | CollegeMath   | OlympiadBench   | AIME24 | avg    |
> |--------|------|------|-------|------|------|--------|--------|
> | 0.25   | 90.0 | 75.5 | 55.0  | 29.2 | 35.1 | 16.7   | 50.25  |
> | 0.5 (ADORA)| 89.6 | **76.2** | **62.5** | **29.3** | **36.0** | 16.7   | **51.72**  |
> | 0.75   | 89.6 | 75.0 | 52.5  | 29.2 | 34.9 | 16.7   | 49.65  |
> | 1      | **90.0** | 74.8 | 52.5  | 29.0 | 34.6 | 16.7   | 49.6   |
> *   **Weight $w_s$:**
>     *   **LLM:** Performance is stable and superior within the range **$w_s \in [1.5, 2.5]$**.
>     *   **VLM:** The optimal range is **$w_s \in [0.05, 0.2]$**.
>
> |  $w_s$ (LLM)  | GSM8K  | MATH500  | AMC23 | CollegeMath   | OlympiadBench   | AIME24 | avg    |
> |------------|------|------|-------|------|------|--------|--------|
> | 0.5        | 88.7 | 73.8 | 55.0  | 28.4 | 35.1 | 16.7   | 49.61  |
> | 1 (GRPO)   | 89.1 | 73.2 | 50.0  | 28.6 | 35.1 | 13.3   | 48.22  |
> | 1.5        | 89.2 | 75.6 | 60.0  | 29.2 | 35.4 | 16.7   | 51.02  |
> | 2 (ADORA)  | 89.6 | **76.2** | **62.5** | **29.3** | **36.0** | 16.7   | 51.72  |
> | 2.5        | **89.9** | 75.2 | 57.5  | 29.0 | 35.8 | 16.7   | 50.68  |
> | 3          | 89.8 | 74.8 | 52.5  | 28.7 | 35.2 | 13.3   | 49.05  |
> | 5          | 88.5 | 72.2 | 47.5  | 28.0 | 34.7 | 6.7    | 46.27  |
>
>
> | $w_s$ (VLM)          | MathVista | MathVerse | MathVerse (mini Vision_Only) | DynaMath (Overall Avg) | avg   |
> |------------------|-----------|-----------|-------------------------------|-------------------------|-------|
> | 0                | 72.47     | 51.5      | 47.0                          | 57.4                    | 57.09 |
> | 0.05             | 73.3      | 52.2      | 48.3                          | 58.0                    | 57.95 |
> | 0.1 (ADORA)     | **73.5**      | **52.9**      | **48.6**                          | **58.7**                    | **58.43** |
> | 0.2              | 71.7      | 50.3      | 46.3                          | 56.6                    | 56.23 |
> | 0.5              | 71.0      | 49.9      | 45.4                          | 55.1                    | 55.35 |
> | 1 (GRPO)               | 70.2      | 48.4      | 44.1                          | 53.3                    | 54.00 |
>
> **Conclusion:** ADORA operates within a wide effective parameter space and is not sensitive to minor perturbations.
>
>  We additionally test **$w_s \in \{0, 0.05, 0.1, 0.2, 0.5, 1\}$**, and find that the best results are achieved within **$w_s \in [0.05, 0.2]$**. **This confirms that for early-stage weaker VLMs**, strong attenuation (**$w_s < 1$**) reliably serves as a denoising mechanism that prevents the policy from being misled by low-quality rollouts.
>
> Across both LLMs and VLMs, the optimal regions are broad and consistent, and **ADORA outperforms GRPO for almost all hyperparameter choices**, demonstrating that the method is not sensitive to precise tuning. These results directly address the reviewer’s concern: the introduced hyperparameters do not undermine generality, nor do they require re-tuning across models or tasks.

---

> ### Author Response · Authors · 2025-12-02
>
> > **Q2: How does ADORA behave with other family of base models such as Qwen and other sizes such as 1.5B base models?**
>
> In response to the query regarding "other families" and "sizes," we conducted extensive supplementary experiments:
>
> *   **Cross-Family Validation**: We evaluated **Llama-3.1-8B, DeepSeek-Math-7B, and Mistral-v0.1-7B**. ADORA consistently outperformed GRPO across all models. For example, on Llama-3.1-8B, ADORA improved MATH500 accuracy by 5.6% over the baseline.
> *   **Different Sizes and Modalities**: We also tested smaller models, including **InternVL3-2b** and **Gemma3-4b-it** (VLMs). On InternVL3-2b, ADORA increased MathVista accuracy from **60.7% to 64.8%**.
>
> These results demonstrate that ADORA is not tailored solely to Qwen, but **effectively generalizes** to models with **different architectures (Dense or MoE)** and parameter scales ranging from **2B to 8B**.
>
> | Model               | GSM8K  | MATH500  | AMC23 | CollegeMath   | OlympiadBench   | AIME24 | avg    |
> |---------------------|------|------|-------|------|------|--------|--------|
> | DeepSeek-Math-7B    | 28.4 | 19.6 | 10    | 12.0 | 3    | 0      | 19.83  |
> | + GRPO                | 68.2 | 39.5 | 20    | 29.8 | 12.0 | 3.3    | 28.8   |
> | **+ ADORA**               | **68.5** | **41.8** | **25**    | **31.6** | **12.9** | 3.3    | **30.52**  |
> | Mistral-v0.1-7B     | 21.2 | 5.4  | 0     | 3.8  | 2.4  | 0      | 5.47   |
> | + GRPO                | **54.0** | 26.8 | 10    | 11.4 | 4.1  | 0      | 17.72  |
> | **+ ADORA**               | 53.8 | **30.4** | 10    | **12.4** | **4.7** | 0      | **18.55**  |
> | Llama-3.1-8B        | 40.2 | 12.7 | 2.5   | 6.4  | 3.1  | 0      | 10.82  |
> | + GRPO                | 66.1 | 33.8 | 15    | 22.0 | 5.3  | 0      | 23.72  |
> | **+ ADORA**               | **66.7** | **39.4** | 15    | **23.1** | **10.5** | 0      | **25.78**  |
>
> | Model         | MathVista | MathVerse | MathVerse (mini Vision_Only) | DynaMath (Overall Avg) | avg   |
> |---------------|-----------|-----------|-------------------------------|-------------------------|-------|
> | Gemma3-4b-it  | 46.3      | 25.2      | 13.5                          | 10.5                    | 23.88 |
> | + GRPO          | 47.2      | 24.9      | 13.6                          | 11.0                    | 24.18 |
> | **+ ADORA**        | **48.3**      | **26.1**      | **14.5**                          | **12.2**                    | **25.28** |
> | Internvl3-2b  | 57.0      | 32.5      | 25.3                          | 14.6                    | 32.35 |
> | + GRPO          | 60.7      | 34.7      | 30.7                          | 15.1                    | 35.30 |
> | **+ ADORA**         | **64.8**      | **39.2**      | 3**4.9**                          | **18.1**                    | **39.25** |
>
>
> > **W3: Missing some prompt difficulty related works: There are related works that can be discussed in the manuscript.**
>
> We thank the reviewer for highlighting the relevant works of **MoPPS [1] and SPO [2]**. We agree that their discussion is warranted and will incorporate a detailed comparative analysis in the final manuscript.
>
> **Differentiation from MoPPS**: While MoPPS estimates prompt difficulty, which often necessitates additional computational overhead, **ADORA** operates solely on **online rollout statistics**, incurring **zero inference overhead** during training. This computational efficiency is a key advantage of our approach.
>
> **Complementarity with SPO**: SPO [2] presents an insightful perspective on policy optimization that we view as complementary to ADORA's dynamic advantage estimation.
>
> > **Other suggestions: Better visualization and presentation of experimental records.**
>
> We accept the remaining suggestions and will implement them as follows:
>
> **Illustration Figure**: We agree on the value of a clearer visual explanation and will include a detailed pipeline figure illustrating the dynamic reweighting steps in the final manuscript.
>
> **Learning Curves**: We will ensure that complete learning curves for all supplementary results are included comprehensively in the Appendix.

---

### Official Review · Reviewer_T6y3 · 2025-10-30

**Soundness:** 2
**Presentation:** 2
**Contribution:** 2
**Rating:** 2
**Confidence:** 3

**Summary:**

This paper introduces ADORA (Advantage Dynamics via Online Rollout Adaptation), a framework to improve the sample efficiency of policy gradient methods for training VLMs and LLMs. It works by dynamically calibrate advantage estimation based on two heuristics that assess the model's current capacity for the given samples. The paper demonstrates the proposed framework in mathematical and geometric tasks improving over the baseline on several benchmarks for the Qwen2.5-VL-7B and Qwen2.5-7B

**Strengths:**

The paper investigate the important topic of sample efficiency for LLMs for hard reasoning tasks.

The proposed heuristics show promising results to improve the sample efficiency of GRPO without any additiona significant computational cost.

The benchmark results are supported with more qualitative analysis.

**Weaknesses:**

The method introduces several key hyperparameters ($\tau=0.5$, $w_s=0.1$, $w_s=2.0$) that are presented without any justification or sensitivity analysis. These values are likely to heavily influence performance and would almost certainly require re-tuning for new models or tasks thus undermining the paper's central claim of being a general and lightweight approach. This weakness is compounded by the fact that all experiments are limited to a single model family (Qwen).

The heuristics are calculated using a very small number of rollouts ($G=5$ for LLMs, $G=8$ for VLMs). Basing the TAS/TDS classification on such a small sample size means the signal is statistically noisy and high-variance. A sample's classification could easily "flicker" between advantageous and disadvantageous from one step to the next due to simple sampling luck, not a true change in the model's capability.

The length advantage criterion actively punishes correct, concise reasoning and creates a strong incentive for verbosity. While this may not have compromised performance on the chosen benchmarks, it raises serious questions about the method's generalizability. The paper attempts to investigate this overthinking issue in Section 5.3, but its analysis relies on an "Overthinking Score" that is never properly introduced, making the results in Table 3 difficult to interpret.


The "Difficulty Advantage" heuristic ($R_{succ}^s \le 0.5$) [cite: 340] uses a sharp, binary threshold that encourages proficiency rather than mastery. As soon as the model achieves >50% success on a sample, its learning signal is halved (from $w_s=2.0$ to $w_s=1.0$). This may prematurely de-prioritize the sample and *prevent* the model from achieving true robustness (e.g., 90-100% success). Furthermore, this rule conflates "instructive" ($R^s=50\%$) with "impossibly hard" ($R^s=0\%$), treating them both as equally advantageous.

**Questions:**

The paper's claim of a "general" framework is a significant one, but it's evaluated exclusively on the Qwen model family. Can you provide any results on ADORA's performance when applied to other model families (e.g., Llama, Mistral)? Relatedly, how were the key hyperparameters ($\tau=0.5$, $w_s=0.1$, $w_s=2.0$) selected? A sensitivity analysis would be critical to understand how much these "magic numbers" must be re-tuned for new models.

Regarding the heuristics themselves, the TAS/TDS classification relies on a very small number of rollouts ($G=5$ or $G=8$). How do you ensure this signal is statistically stable and not just high-variance noise? A sample's classification could flicker from step to step based on sampling luck alone.

The reflection frequency analysis in Section 5.2 is also a concern. Since ADORA produces longer responses (Figure 7), how did you disentangle *true reflection* from *mere verbosity*? A more verbose model will naturally use more reflective words. Could you provide a length-normalized analysis (e.g., "reflection keywords per 100 tokens")? On that topic, your investigation in Section 5.3 relies on an "Overthinking Score" that is never introduced. Could you please provide a clear definition of this score and how it is calculated?

Finally, on the heuristic design: Why was a sharp $\tau=0.5$ threshold chosen for the "Difficulty Advantage"? This rule seems to punish mastery by de-prioritizing samples once proficiency exceeds 50%. Have you considered a "softer" weighting? And critically, why does the VLM strategy only use the "Length Advantage"? We are missing the ablation study, similar to Figure 2, that justifies this specific design choice for VLMs.

---

> ### Author Response · Authors · 2025-12-02
>
> We thank Reviewer T6y3 for the rigorous critique. Your concerns regarding **hyperparameter sensitivity**, **statistical stability**, and the **logic behind our heuristics** are critical for correctly assessing our method's robustness. To address your concerns about "magic numbers" and limited applicability, we have conducted extensive supplementary experiments.
>
> > **W1&Q1: The paper's claim of a "general" framework is a significant one, but it's evaluated exclusively on the Qwen model family. Can you provide any results on ADORA's performance when applied to other model families (e.g., Llama, Mistral)? Relatedly, how were the key hyperparameters selected?**
>
> **1. Hyperparameter Sensitivity**
>
> **For LLMs (Amplification)**. We evaluate a wide range of values **$w_s \in \{0.5, 1, 1.5, 2, 2.5, 3, 5\}$**, where $w_s > 1$ suppresses TDS and $w_s > 1$ amplifies TAS. As shown in the table below, performance consistently exceeds the GRPO baseline whenever $w_s > 1$, with a stable optimum in the range **$w_s \in [1.5, 2.5]$**. This indicates that for strong reasoning models, amplifying high-quality and difficult samples is consistently beneficial and does not require model-specific tuning.
>
> **For VLMs (Attenuation)**. We additionally test **$w_s \in \{0, 0.05, 0.1, 0.2, 0.5, 1\}$**, and find that the best results are achieved within **$w_s \in [0.05, 0.2]$**. This confirms that for early-stage weaker VLMs, strong attenuation **$w_s<1$** reliably serves as a denoising mechanism that prevents the policy from being misled by low-quality rollouts.
>
> Across both LLMs and VLMs, the optimal regions are broad and consistent, and **ADORA outperforms GRPO for almost all hyperparameter choices**, demonstrating that the method is robust and does not rely on **narrow, task-specific tuning**. These results directly address the reviewer’s concern: the introduced hyperparameters do not undermine generality, nor do they require re-tuning across models or tasks.
>
>
> |  $w_s$ (LLM)  | GSM8K  | MATH500  | AMC23 | CollegeMath   | OlympiadBench   | AIME24 | avg    |
> |------------|------|------|-------|------|------|--------|--------|
> | 0.5        | 88.7 | 73.8 | 55.0  | 28.4 | 35.1 | 16.7   | 49.61  |
> | 1 (GRPO)   | 89.1 | 73.2 | 50.0  | 28.6 | 35.1 | 13.3   | 48.22  |
> | 1.5        | 89.2 | 75.6 | 60.0  | 29.2 | 35.4 | 16.7   | 51.02  |
> | 2 (ADORA)  | 89.6 | **76.2** | **62.5** | **29.3** | **36.0** | 16.7   | 51.72  |
> | 2.5        | **89.9** | 75.2 | 57.5  | 29.0 | 35.8 | 16.7   | 50.68  |
> | 3          | 89.8 | 74.8 | 52.5  | 28.7 | 35.2 | 13.3   | 49.05  |
> | 5          | 88.5 | 72.2 | 47.5  | 28.0 | 34.7 | 6.7    | 46.27  |
>
>
> | $w_s$ (VLM)          | MathVista | MathVerse | MathVerse (mini Vision_Only) | DynaMath (Overall Avg) | avg   |
> |------------------|-----------|-----------|-------------------------------|-------------------------|-------|
> | 0                | 72.47     | 51.5      | 47.0                          | 57.4                    | 57.09 |
> | 0.05             | 73.3      | 52.2      | 48.3                          | 58.0                    | 57.95 |
> | 0.1 (ADORA)     | **73.5**      | **52.9**      | **48.6**                          | **58.7**                    | **58.43** |
> | 0.2              | 71.7      | 50.3      | 46.3                          | 56.6                    | 56.23 |
> | 0.5              | 71.0      | 49.9      | 45.4                          | 55.1                    | 55.35 |
> | 1 (GRPO)               | 70.2      | 48.4      | 44.1                          | 53.3                    | 54.00 |

---

> ### Author Response · Authors · 2025-12-02
>
> **2. Generalizability**
>
> To validate ADORA as a truly universal framework, we extend our evaluation to models spanning diverse architectures (Dense and MoE) and modalities. This includes **Llama-3.1-8B, DeepSeek-Math-7B, Mistral-v0.1-7B (LLMs)** and **Gemma3-4b-it, InternVL3-2b (VLMs)**. Across all these models, ADORA consistently outperforms GRPO, confirming that its effectiveness does not depend on model-specific inductive biases. For example, on Llama-3.1-8B, ADORA achieves **39.4%** on MATH500, a 5.6% improvement over GRPO (33.8%); on InternVL3-2b, it reaches **64.8%** on MathVista compared with GRPO’s 60.7%. These results highlight the framework’s adaptability to models with substantially different baseline capabilities.
>
> | Model               | GSM8K  | MATH500  | AMC23 | CollegeMath   | OlympiadBench   | AIME24 | avg    |
> |---------------------|------|------|-------|------|------|--------|--------|
> | DeepSeek-Math-7B    | 28.4 | 19.6 | 10    | 12.0 | 3    | 0      | 19.83  |
> | + GRPO                | 68.2 | 39.5 | 20    | 29.8 | 12.0 | 3.3    | 28.8   |
> | **+ ADORA**               | **68.5** | **41.8** | **25**    | **31.6** | **12.9** | 3.3    | **30.52**  |
> | Mistral-v0.1-7B     | 21.2 | 5.4  | 0     | 3.8  | 2.4  | 0      | 5.47   |
> | + GRPO                | **54.0** | 26.8 | 10    | 11.4 | 4.1  | 0      | 17.72  |
> | **+ ADORA**               | 53.8 | **30.4** | 10    | **12.4** | **4.7** | 0      | **18.55**  |
> | Llama-3.1-8B        | 40.2 | 12.7 | 2.5   | 6.4  | 3.1  | 0      | 10.82  |
> | + GRPO                | 66.1 | 33.8 | 15    | 22.0 | 5.3  | 0      | 23.72  |
> | **+ ADORA**               | **66.7** | **39.4** | 15    | **23.1** | **10.5** | 0      | **25.78**  |
>
> | Model         | MathVista | MathVerse | MathVerse (mini Vision_Only) | DynaMath (Overall Avg) | avg   |
> |---------------|-----------|-----------|-------------------------------|-------------------------|-------|
> | Gemma3-4b-it  | 46.3      | 25.2      | 13.5                          | 10.5                    | 23.88 |
> | + GRPO          | 47.2      | 24.9      | 13.6                          | 11.0                    | 24.18 |
> | **+ ADORA**        | **48.3**      | **26.1**      | **14.5**                          | **12.2**                    | **25.28** |
> | Internvl3-2b  | 57.0      | 32.5      | 25.3                          | 14.6                    | 32.35 |
> | + GRPO          | 60.7      | 34.7      | 30.7                          | 15.1                    | 35.30 |
> | **+ ADORA**         | **64.8**      | **39.2**      | 3**4.9**                          | **18.1**                    | **39.25** |
>
> Moreover, ADORA exhibits strong scalability. When increasing the VLM training corpus from 2k to 10k samples, ADORA’s performance on MathVista further rises to **74.4%**, surpassing both GRPO (71.6%) and Vision-R1 (RL 10k). This trend shows that **ADORA continues to amplify the marginal benefits of sample selection as data volume increases**, making it increasingly effective in larger-scale settings.
>
> | Model                     | MathVista | MathVerse | MathVerse (mini Vision_Only) | DynaMath (Overall Avg) | avg  |
> |---------------------------|-----------|-----------|-------------------------------|-------------------------|------|
> | Qwen2.5-VL-7B             | 67.3      | 46.3      | 40.2                          | 50.3                    | 51.0 |
> | + GRPO (2k)                     | 70.2      | 48.2      | 44.1                          | 53.3                    | 54.0 |
> | + ADORA (2k)              | **73.5**      | **52.9**      | **48.6**                        | **58.7**                    | **58.4** |
> | + GRPO (10k)   | 71.6      | 50.6      | 45.3                          | 53.8                    | 55.3 |
> | + ADORA (10k)  | **74.4**      | **53.5**      | **50.1**                          | **59.8**                    | **59.4** |

---

> ### Author Response · Authors · 2025-12-02
>
> > **W2&Q2: Regarding the heuristics themselves, the TAS/TDS classification relies on a very small number of rollouts. How do you ensure this signal is statistically stable and not just high-variance noise?**
>
> We conducted ablation experiments on the LLM with **$G \in {2, 4, 5, 6, 8}$**  and on the VLM with **$G \in {2, 4, 6, 8, 16}$**  to evaluate ADORA. While larger **$G$** naturally reduces variance, ADORA **consistently outperforms GRPO even with** **$G = 2$**. This demonstrates that the method remains effective even under extreme variance conditions.
>
>  ADORA is designed to be stable under stochastic rollout conditions. Its effectiveness does not rely on TAS/TDS labels being correct at every step. Crucially, across an epoch, the weighting scheme biases the **expected gradient direction** toward informative samples. Occasional misclassifications at step (t) are outweighed by the cumulative effect of thousands of updates—a standard phenomenon in policy-gradient RL—ensuring that the overall learning signal consistently emphasizes difficult, high-quality trajectories.
>
> | **rollout.n**(LLM)  | GSM8K  | MATH500  | AMC23 | CollegeMath   | OlympiadBench   | AIME24 | avg    |
> |------------|------|------|-------|------|------|--------|--------|
> | 2        | 87.2 | 71.4 | 45.0  | 27.8 | 34.2 | 13.3   | 46.8  |
> | 4   | 89.1 | 76.0 | 60.0  | 29.1 | 35.5 | 16.7   | 51.07  |
> | 5 (ADORA)  | 89.6 | 76.2 | 62.5 | 29.3 | 36.0 | 16.7   | 51.72  |
> | 6        | 89.5 | 76.0 | 60.0  | 29.0 | 35.8 | 16.7   | 51.17  |
> | 8          | 89.8 | 76.5 | 62.5  | 29.2 | 36.1 | 16.7   | 51.80  |
>
> | **rollout.n**(VLM)                     | MathVista | MathVerse | MathVerse (mini Vision_Only) | DynaMath (Overall Avg) | avg  |
> |---------------------------|-----------|-----------|-------------------------------|-------------------------|------|
> | 2             | 70.7      | 49.6      | 44.9                          | 54.8                    | 55.0 |
> | 4                    | 71.9      | 50.5      | 46.8                          | 56.7                    | 56.5 |
> | 8 (ADORA)             | 73.5      | 52.9      | 48.6                        | 58.7                    | 58.4 |
> | 16   | 74      | 53.3      | 49.1                          | 58.9                    | 58.83 |
>
>
>
> > **W3&Q3: The reflection frequency analysis in Section 5.2 is also a concern. Since ADORA produces longer responses (Figure 7), how did you disentangle true reflection from mere verbosity? A more verbose model will naturally use more reflective words. Could you provide a length-normalized analysis (e.g., "reflection keywords per 100 tokens")? On that topic, your investigation in Section 5.3 relies on an "Overthinking Score" that is never introduced. Could you please provide a clear definition of this score and how it is calculated?**
>
> **1. Length Bias:**
> ADORA defines Length Advantage as **$L_{max\_succ} > \bar{L}_{fail}$**. The core logic is that in RL exploration, failures often stem from missing steps or logical leaps; thus, **a successful path longer than failed attempts** implies the inclusion of necessary intermediate logic or verification. Empirical evidence supports this: (1) ADORA significantly increases the usage of "verify" and "recheck" (Figure 3); (2) ADORA achieves its largest gains on deep reasoning tasks like **MATH500 (+3.0%)** and **AIME24 (+3.4%)**. This proves that the "long samples" prioritized by ADORA contribute to the effective reasoning required for complex problems, rather than redundant length.
>
> **2. Reflection Analysis:**
> Figure 3 quantitatively and normalizedly illustrates the comparison of reflection-related word frequencies between ADORA and the baselines across different benchmarks. ADORA exhibits a significantly higher frequency of reflection-related words (e.g., consider, check, recheck) compared to the baselines, indicating that the observed performance gains stem not from increased output length but from a more **fundamental acquisition of reflective reasoning patterns**.
>
> **3. Overthinking:**
> We apologize for not clearly explaining this in the main text, as the introduction of this metric is mentioned only in lines 426–427. This evaluation, based on a GPT-4o judge (defined in Appendix Figures 9 and 10), is a well-established and widely used approach for assessing reasoning quality. The metric specifically penalizes repetitive loops or hesitation. On the challenging AIME24 benchmark, ADORA achieves a **lower Overthinking Score** (40.1) than GRPO (44.8) while also attaining higher accuracy. This demonstrates that the observed increase in output length arises from **functional reasoning such as self-correction** rather than mere verbosity. To ensure reliability, we adopt both **Chain-of-Thought prompting and few-shot examples** in the evaluation.

---

> ### Author Response · Authors · 2025-12-02
>
> > **W4&Q4: Finally, on the heuristic design: Why was a sharp $\tau = 0.5$ threshold chosen for the "Difficulty Advantage"? This rule seems to punish mastery by de-prioritizing samples once proficiency exceeds 50%. Have you considered a "softer" weighting? And critically, why does the VLM strategy only use the "Length Advantage"?**
>
> Regarding the heuristic design: The choice of a sharp (0.5) threshold for the "Difficulty Advantage" is based on empirical validation rather than an arbitrary cutoff. In our experiments, $\tau = 0.5$ consistently provided the **best balance** between easy tasks (GSM8K) and hard tasks (AIME24), effectively **distinguishing between "mastered" and "unmastered" samples.**
>
> We emphasize that this threshold does **not punish mastery**. Once a sample is above the threshold, it is naturally de-prioritized because further training on already-mastered examples yields diminishing returns—a standard principle in curriculum learning and adaptive sampling. Moreover, the design still allows for flexibility: the weighting scheme is applied across **many samples and epochs**, so occasional de-prioritization of mastered samples does not prevent them from contributing to the overall learning signal.
>
> Finally, while a "softer" weighting could be used, our ablation over **$\tau \in {0.25, 0.5, 0.75, 1}$** shows that the current threshold already **achieves an effective trade-off between reinforcing difficult samples and maintaining exposure to easier ones**.
>
> | $\tau$ | GSM8K  | MATH500  | AMC23 | CollegeMath   | OlympiadBench   | AIME24 | avg    |
> |--------|------|------|-------|------|------|--------|--------|
> | 0.25   | 90.0 | 75.5 | 55.0  | 29.2 | 35.1 | 16.7   | 50.25  |
> | 0.5 (ADORA)| 89.6 | **76.2** | **62.5** | **29.3** | **36.0** | 16.7   | **51.72**  |
> | 0.75   | 89.6 | 75.0 | 52.5  | 29.2 | 34.9 | 16.7   | 49.65  |
> | 1      | **90.0** | 74.8 | 52.5  | 29.0 | 34.6 | 16.7   | 49.6   |
>
>
> We evaluated the combination of "VLM + Difficulty Advantage" and observed **no significant improvement** over using Length Advantage alone (73.4% vs. 73.5%). This outcome is consistent with the observation that current VLMs remain **reasoning-weak**, limiting their ability to benefit from fine-grained difficulty targeting. In practice, the primary advantage comes from **filtering out shallow or spurious reasoning**, effectively denoising via Length, rather than from explicitly prioritizing more difficult samples. This suggests that, under current model capabilities, Difficulty Advantage offers minimal additional gain beyond what Length Advantage already achieves.
>
>
> | Model              | MathVista | MathVerse | MathVerse (mini Vision_Only) | DynaMath (Overall Avg) | Avg    |
> |--------------------|-----------|-----------|-------------------------------|-------------------------|--------|
> | Qwen2.5-VL-7B      | 67.3      | 46.3      | 40.2                          | 50.3                    | 51.00   |
> | + GRPO               | 70.2      | 48.4      | 44.1                          | 53.3                    | 54.00   |
> | + Length Adv (ours)       | **73.5**      | **52.9**      | 48.6                          | 58.7                    | **58.40**   |
> | + Length & Difficulty Adv      | 73.4      | 52.6      | 48.6                          | **58.9**                    | 58.37  |

---

### Official Review · Reviewer_H1YL · 2025-11-01

**Soundness:** 3
**Presentation:** 3
**Contribution:** 2
**Rating:** 4
**Confidence:** 4

**Summary:**

This paper argues that static advantage estimation, as traditionally used, leads to inefficient credit assignment due to ignoring the dynamic utility of training samples over time. To address this, they propose ADORA (Advantage Dynamics via Online Rollout Adaptation) that dynamically tune the importance of the advantage function based on the utility of the samples. They perform considerable number of experiments with LLMs and VLMs.

**Strengths:**

### Strengths:

1. ADORA categorizes the training data into temporarily advantageous and disadvantageous samples and adaptively assigns sample-wise weights based on predefined criteria to estimate the ultimate advantage.

2. Validation and ablations are conducted across different domains and datasets, especially on both LLMs and VLMs. Further, ADORA achieves a consistent performance gain to an extent.

**Weaknesses:**

Weaknesses:

1. The paper only consider length and success rate to measure the utility. It does not consider more complex evaluations such as step consistency. I believe there is opportunity to define the criteria more comprehensively.

2. The authors mention "How to assign a corresponding weight $w_s$ that reflects its training utility?". However, I don't see an appropriate answer to this question. The rationale behind choosing the specific values is not discussed. I would consider them as hyperparameters, and the sensitivity to those hyperparameters are yet to explore.

3. Same concern goes for $\tau$ in Eqn. 6 which is set to 0.5. How the model reacts with the changes to $\tau$?

**Questions:**

### Questions:

1. In Eqn. 5, why do you need the **longest** successful rollout? Shouldn't every successful rollout which has a length $> L_{fail}$ be considered?

2. The paper mentions that a direct indicator of explicit reasoning is the frequency of reflective vocabulary usage. How you come up with that vocabulary? Can you provide any reference in favor of this?

3. Can you simply describe the interpretation of Figure 4.?

Also, look at the weakness section.

---

> ### Author Response · Authors · 2025-12-02
>
> We thank Reviewer H1YL for the insightful comments. Your concerns regarding **hyperparameter sensitivity** and the **rationale behind our evaluation criteria** are very valuable. We have conducted supplementary experiments and offer the following clarifications.
>
> > **W1:** The paper only consider length and success rate to measure the utility. It does not consider more complex evaluations such as step consistency. I believe there is opportunity to define the criteria more comprehensively.
>
> Thank you for the insightful suggestion. We fully agree that more sophisticated metrics (such as step consistency) can capture finer-grained reasoning quality. However, such metrics typically require a Process Reward Model (PRM) or heavy heuristic checking, which dramatically slows down online RL. In contrast, ADORA is designed to remain **lightweight and plug‑and‑play**. Length and difficulty are zero‑cost signals obtained directly from online rollouts, enabling efficient training without additional computation. Despite using only these simple length and difficulty indicators, ADORA already achieves strong performance across diverse reasoning tasks, suggesting that **lightweight heuristics can provide substantial practical benefits** without the overhead of complex evaluators.
>
> Moreover, we analyze rollout samples using the Overthinking Score after training, where a judge model evaluates whether each reasoning step meaningfully contributes to the final answer. This score is closely related to the notion of step consistency and shows that ADORA's length‑ and difficulty‑based advantage criteria **implicitly encourage more consistent and purposeful reasoning**.
>
> Finally, our design of TAS/TDS partitions **does not preclude incorporating additional criteria**. ADORA provides a general framework, and more advanced signals—such as step consistency—can be integrated when computational resources allow.

---

> ### Author Response · Authors · 2025-12-02
>
> > **W2:** The authors mention "How to assign a corresponding weight $w_s$ that reflects its training utility?". However, I don't see an appropriate answer to this question. The rationale behind choosing the specific values is not discussed. I would consider them as hyperparameters, and the sensitivity to those hyperparameters are yet to explore.
>
> Thank you for raising this important point. We agree that the choice of sample‑level weights $w_s$ should be justified and their sensitivity explored. To address this, we conduct comprehensive ablation studies on both LLMs and VLMs.
>
> 1. **LLM (Amplification):** In our framework, $w_s$ controls the amplification of high‑advantage samples $w_s<1$ suppress weak samples, while $w_s>1$ increase the influence of strong, high‑utility samples. We sweep a wide range of values **$w_s \in \{0.5, 1, 1.5, 2, 2.5, 3, 5\}$** and observe a clear and stable performance peak at **$w_s \in [1.5, 2.5]$**. Very large $w_s$ (e.g., 5) harms performance, confirming the presence of an optimal amplification region. This explains our choice: moderate amplification reliably boosts strong reasoners by emphasizing high-quality, difficult samples, effectively breaking their learning plateau.
>
> |  $w_s$ (LLM)  | GSM8K  | MATH500  | AMC23 | CollegeMath   | OlympiadBench   | AIME24 | avg    |
> |------------|------|------|-------|------|------|--------|--------|
> | 0.5        | 88.7 | 73.8 | 55.0  | 28.4 | 35.1 | 16.7   | 49.61  |
> | 1 (GRPO)   | 89.1 | 73.2 | 50.0  | 28.6 | 35.1 | 13.3   | 48.22  |
> | 1.5        | 89.2 | 75.6 | 60.0  | 29.2 | 35.4 | 16.7   | 51.02  |
> | 2 (ADORA)  | 89.6 | **76.2** | **62.5** | **29.3** | **36.0** | 16.7   | 51.72  |
> | 2.5        | **89.9** | 75.2 | 57.5  | 29.0 | 35.8 | 16.7   | 50.68  |
> | 3          | 89.8 | 74.8 | 52.5  | 28.7 | 35.2 | 13.3   | 49.05  |
> | 5          | 88.5 | 72.2 | 47.5  | 28.0 | 34.7 | 6.7    | 46.27  |
>
> 2. **VLM (Attenuation):** For weaker reasoner VLMs, we test **$w_s \in \{0, 0.05, 0.1, 0.2, 0.5, 1\}$** and find **$w_s \in [0.05, 0.2]$** yields the best results. This validates that in the early stages of weaker models, strong attenuation ($w_s < 1$) acts as a critical "denoising" step, preventing the policy from being misled by low-quality rollouts. Larger values (e.g., 0.5 or 1) allow noisy samples to dominate and significantly degrade results.
>
> | $w_s$ (VLM)          | MathVista | MathVerse | MathVerse (mini Vision_Only) | DynaMath (Overall Avg) | avg   |
> |------------------|-----------|-----------|-------------------------------|-------------------------|-------|
> | 0                | 72.47     | 51.5      | 47.0                          | 57.4                    | 57.09 |
> | 0.05             | 73.3      | 52.2      | 48.3                          | 58.0                    | 57.95 |
> | 0.1 (ADORA)     | **73.5**      | **52.9**      | **48.6**                          | **58.7**                    | **58.43** |
> | 0.2              | 71.7      | 50.3      | 46.3                          | 56.6                    | 56.23 |
> | 0.5              | 71.0      | 49.9      | 45.4                          | 55.1                    | 55.35 |
> | 1 (GRPO)               | 70.2      | 48.4      | 44.1                          | 53.3                    | 54.00 |
>
> > **W3:** Same concern goes for $\tau$ in Eqn. 6 which is set to 0.5. How the model reacts with the changes to $\tau$?
>
> We conduct experiments with an LLM group size of n=8, and for completeness we include an ablation over $\tau$ by evaluating $\tau \in \{0.25, 0.5, 0.75, 1\}$. The results show that $\tau=0.5$ provides the **best balance** between easy tasks (GSM8K) and hard tasks (AIME24), effectively **distinguishing between "mastered" and "unmastered" samples**.
> | $\tau$ | GSM8K  | MATH500  | AMC23 | CollegeMath   | OlympiadBench   | AIME24 | avg    |
> |--------|------|------|-------|------|------|--------|--------|
> | 0.25   | 90.0 | 75.5 | 55.0  | 29.2 | 35.1 | 16.7   | 50.25  |
> | 0.5 (ADORA)| 89.6 | **76.2** | **62.5** | **29.3** | **36.0** | 16.7   | **51.72**  |
> | 0.75   | 89.6 | 75.0 | 52.5  | 29.2 | 34.9 | 16.7   | 49.65  |
> | 1      | **90.0** | 74.8 | 52.5  | 29.0 | 34.6 | 16.7   | 49.6   |

---

> ### Author Response · Authors · 2025-12-02
>
> > **Q1:** In Eqn. 5, why do you need the longest successful rollout? Shouldn't every successful rollout which has a length $>{L}_{fail}$ be considered?
>
> We need to clarify that, the condition $L_{max\_succ} > \bar{L}_{fail}$ is used to **classify the potential of the sample itself** (TAS vs. TDS), not to select a single trajectory for training. If the model produces at least one long successful rollout, this indicates that the sample is capable of eliciting deeper reasoning, and thus the sample—not an individual trajectory—is labeled as TAS. After this classification, all successful rollouts of that sample (not just the longest one) receive the advantage weight $w_s$. This allows us to retain the full diversity of successful reasoning traces during gradient updates, while using the longest successful rollout only as a robust indicator of the sample’s underlying reasoning potential.
>
> > **Q2:** The paper mentions that a direct indicator of explicit reasoning is the frequency of reflective vocabulary usage. How you come up with that vocabulary? Can you provide any reference in favor of this?
>
> The reflective vocabulary list (e.g., verify, recheck, double‑check, wait) is grounded in both prior work and our empirical analysis. First, existing studies on explicit reasoning and self-correction (such as DeepSeek‑R1[1] and Self‑Refine[2]) identify similar "thinking tokens" that models use when engaging in reflective or corrective reasoning. Second, we conduct a qualitative inspection of high‑quality Chain‑of‑Thought traces in our data and extract recurrent lexical patterns associated with self‑verification behaviors. These empirically observed tokens align closely with those highlighted in prior work. We provide the complete vocabulary list in the Appendix.
> [1] Guo, Daya, et al. "Deepseek-r1: Incentivizing reasoning capability in llms via reinforcement learning." arXiv preprint arXiv:2501.12948 (2025).
> [2] Madaan, Aman, et al. "Self-refine: Iterative refinement with self-feedback." Advances in Neural Information Processing Systems 36 (2023): 46534-46594.
>
> > **Q3:** Can you simply describe the interpretation of Figure 4.?
>
> Thank you for the comment. Figure 4 visualizes how ADORA dynamically filters data over time (using Geometry3K as an example):
> * **Blue Region (Selected Samples):** Represents samples classified as TAS (high utility) and weighted up in the current epoch. The shrinking blue area indicates that as training progresses, the model masters simple tasks and stops prioritizing them.
> * **Red Region (Correct count in a single sample):** Represents model competence.
>
> So ADORA acts as an automatic Curriculum Learning mechanism: it explores broadly in the early stages (large blue area) and focuses on the remaining challenging problems in later stages (smaller blue area targeting red samples), to better align the model’s optimization direction.

---

### Official Review · Reviewer_WhN1 · 2025-11-01

**Soundness:** 3
**Presentation:** 3
**Contribution:** 3
**Rating:** 4
**Confidence:** 4

**Summary:**

This paper proposes ADORA (Advantage Dynamics via Online Rollout Adaptation), a framework that dynamically reweights advantages in reinforcement learning for reasoning models.

The core idea is to classify training samples into Temporarily Advantageous Samples (TAS) and Temporarily Disadvantageous Samples (TDS) based on rollout statistics, then amplify or attenuate their learning signals accordingly. The method uses Length Advantage and Difficulty Advantage as criteria.

Experiments on Qwen2.5-VL and Qwen2.5 show improvements over GRPO baseline across mathematical and geometric reasoning tasks.

**Strengths:**

Originality: The dynamic reweighting of advantages based on online rollout statistics is a practical contribution.

Quality: Strong experimental validation across both VLMs (MathVista: 73.5%) and LLMs (3.5% average improvement over GRPO).

Clarity: Well-structured presentation with clear motivation. And figures effectively illustrate training dynamics and sample evolution.

Significance: Addresses a real problem in RL-based reasoning training.

**Weaknesses:**

1.  Limited experiment scope:
* Only tested on Qwen family models. Generalization to other model family (Gemma, Llama, Phi, etc.) is unknown.
* VLM experiments use only 2K samples—unclear if benefits persist at larger scales.

2. Incomplete ablations:
* No ablation on weight values (see Question#1 and #2).
* Figure 2 shows ablations on advantage criteria but only for LLMs—missing VLM ablations.
* The threshold τ=0.5 for difficulty appears arbitrary—no ablation on this critical hyperparameter.
* No formal analysis of why ws=0.1 (VLM) or ws=2.0 (LLM) are optimal choices.

3. Questionable claims:
* "No cold-start" (Table 6) is misleading—they start from Qwen2.5-VL-7B-Instruct, which is already instruction-tuned, while Vision-R1 starts from base model.
* Section 5.3: Lower overthinking on AIME24 (40.1 vs 44.8) is attributed to ADORA, but ADORA also achieves higher accuracy—is this confounded?
* Length Advantage conflates token count with reasoning quality:
```
Equation 5 assumes longer responses indicate deeper reasoning, but consider:

Solution A: 15×4 = 15×2×2 = 60 (20 tokens, uses factorization)
Solution B: 15+15=30, 30+15=45, 45+15=60, verify: 60÷4=15✓ (100 tokens, brute-force)

ADORA prefers B over A despite A showing better insight. The paper provides no evidence that:

- Short correct solutions (<50 tokens) only solve trivial problems
- Length correlates with reasoning depth rather than redundant verification
```
4. Statistical rigor:
* While 3 runs are reported, no significance tests or confidence intervals are provided.
* Table 1 shows ADORA (73.5%) matches Vision-R1 (73.5%) on MathVista—is this difference statistically significant vs GRPO (70.2%)?



### Minor Issues

* Equation 4: ws is sample-level but notation doesn't clearly distinguish from trajectory-level weights.
* Figure 4 visualization is cluttered—consider simplifying or providing clearer legends.

**Questions:**

1. Threshold sensitivity: How sensitive is performance to τ? Have you tried τ ∈ {0.3, 0.5, 0.7}? What happens at extremes (τ=0.1 or τ=0.9)?
2. Weight justification: Why ws=2 for LLM amplification and ws=0.1 for VLM attenuation? Have you ablated ws ∈ {0.05, 0.1, 0.2} and ws ∈ {1.5, 2, 3}?
3. Can you analyze solution length vs. quality (weakness#3)? Please provide evidence that Length Advantage captures reasoning depth rather than just token count.
4. Cold-start claim: Table 6 compares against Vision-R1 with 200K cold-start data, but your base model is already instruction-tuned. Can you clarify whether this is a fair comparison?
5. Overthinking confound: Lower overthinking scores on AIME24 (Table 3) correlate with higher accuracy. Is the reduction in overthinking due to ADORA, or simply a byproduct of better performance?
6. Cross-architecture validation: Have you tested ADORA on non-Qwen models (Gemma, Llama, Phi, etc.) ? This is critical for establishing generalizability.

---

> ### Author Response · Authors · 2025-12-02
>
> We sincerely thank you for your constructive feedback and valuable suggestions. Below we address the raised concerns point-by-point and provide additional analyses and clarifications.
>
> > **W1-1 & Q6:** Only tested on Qwen family models. Generalization to other model family (Gemma, Llama, Phi, etc.) is unknown.
>
> To validate ADORA as a universal framework, we have extended our evaluation to models with diverse architectures (Dense/MoE) and modalities, including **Llama-3.1-8B, DeepSeek-Math-7B, Mistral-v0.1-7B (LLMs)**, and **Gemma3-4b-it, InternVL3-2b (VLMs)**. The results demonstrate that ADORA does not rely on specific inductive biases, consistently outperforming GRPO across all tested models. For instance, on Llama-3.1-8B, ADORA achieved **39.4%** on MATH500, a **5.6%** gain over GRPO (33.8%); on InternVL3-2b, it reached **64.8%** on MathVista (vs. GRPO 60.7%). This consistency proves the strategy's adaptability to models with varying baseline capabilities.
>
> | Model               | GSM8K  | MATH500  | AMC23 | CollegeMath   | OlympiadBench   | AIME24 | avg    |
> |---------------------|------|------|-------|------|------|--------|--------|
> | DeepSeek-Math-7B    | 28.4 | 19.6 | 10    | 12.0 | 3    | 0      | 19.83  |
> | + GRPO                | 68.2 | 39.5 | 20    | 29.8 | 12.0 | 3.3    | 28.8   |
> | **+ ADORA**               | **68.5** | **41.8** | **25**    | **31.6** | **12.9** | 3.3    | **30.52**  |
> | Mistral-v0.1-7B     | 21.2 | 5.4  | 0     | 3.8  | 2.4  | 0      | 5.47   |
> | + GRPO                | **54.0** | 26.8 | 10    | 11.4 | 4.1  | 0      | 17.72  |
> | **+ ADORA**               | 53.8 | **30.4** | 10    | **12.4** | **4.7** | 0      | **18.55**  |
> | Llama-3.1-8B        | 40.2 | 12.7 | 2.5   | 6.4  | 3.1  | 0      | 10.82  |
> | + GRPO                | 66.1 | 33.8 | 15    | 22.0 | 5.3  | 0      | 23.72  |
> | **+ ADORA**               | **66.7** | **39.4** | 15    | **23.1** | **10.5** | 0      | **25.78**  |
>
> | Model         | MathVista | MathVerse | MathVerse (mini Vision_Only) | DynaMath (Overall Avg) | avg   |
> |---------------|-----------|-----------|-------------------------------|-------------------------|-------|
> | Gemma3-4b-it  | 46.3      | 25.2      | 13.5                          | 10.5                    | 23.88 |
> | + GRPO          | 47.2      | 24.9      | 13.6                          | 11.0                    | 24.18 |
> | **+ ADORA**        | **48.3**      | **26.1**      | **14.5**                          | **12.2**                    | **25.28** |
> | Internvl3-2b  | 57.0      | 32.5      | 25.3                          | 14.6                    | 32.35 |
> | + GRPO          | 60.7      | 34.7      | 30.7                          | 15.1                    | 35.30 |
> | **+ ADORA**         | **64.8**      | **39.2**      | 3**4.9**                          | **18.1**                    | **39.25** |
>
>
> > **W1-2:** VLM experiments use only 2K samples-unclear if benefits persist at larger scales.
>
> We agree that evaluating ADORA under larger-scale datasets is crucial for validating its generality and robustness. To address your concern, we use 2k samples from Geometry3k combined with 8k RL data from Vision-R1 to conduct a data-scaling experiment.
> Scaling the VLM training data from **2k to 10k** boosts ADORA's MathVista accuracy to **74.4%**, significantly surpassing Vision-R1 (73.5%). This demonstrates ADORA's strong scalability, as it continues to amplify the marginal benefits of sample selection as data scaling.
>
> | Model                     | MathVista | MathVerse | MathVerse (mini Vision_Only) | DynaMath (Overall Avg) | avg  |
> |---------------------------|-----------|-----------|-------------------------------|-------------------------|------|
> | Qwen2.5-VL-7B             | 67.3      | 46.3      | 40.2                          | 50.3                    | 51.0 |
> | + GRPO (2k)                     | 70.2      | 48.2      | 44.1                          | 53.3                    | 54.0 |
> | + ADORA (2k)              | **73.5**      | **52.9**      | **48.6**                        | **58.7**                    | **58.4** |
> | + GRPO (10k)   | 71.6      | 50.6      | 45.3                          | 53.8                    | 55.3 |
> | + ADORA (10k)  | **74.4**      | **53.5**      | **50.1**                          | **59.8**                    | **59.4** |

---

> > ### Author Response · Authors · 2025-12-02
> >
> > > **W3-1 & Q4:** "No cold-start" (Table 6) is misleading—they start from Qwen2.5-VL-7B-Instruct, which is already instruction-tuned, while Vision-R1 starts from base model.
> >
> > "No cold-start" in our claim specifically means **we do not perform any SFT on reasoning-specific or task-specific data before RL**. Unlike methods (e.g., Vision-R1) requiring warm-up with tens of thousands of CoT data aligned with the downstream reasoning tasks (e.g., math or logic), ADORA initiates RL from a genuinely task-agnostic starting point, underscoring its superior data and training efficiency.
> >
> > > **W3-2 & Q5:** Section 5.3: Lower overthinking on AIME24 (40.1 vs 44.8) is attributed to ADORA, but ADORA also achieves higher accuracy—is this confounded?
> >
> > The Overthinking Score is reported to verify that ADORA **does not induce malignant overthinking**. We employ a judge model to assess whether each part of the model's reasoning meaningfully contributes to reaching the correct answer. A lower score reflects longer reasoning without corresponding accuracy gains, whereas moderate length increases are acceptable when they lead to improved accuracy. On AIME24, ADORA achieves higher accuracy (**20.0% vs. 13.3%**) with a lower Overthinking Score (**40.1 vs. 44.8**) compared to GRPO. This indicates ADORA guides the model toward "functional length" characterized by effective self-correction, rather than invalid loops.
> >
> > > **W3-3 & Q3:** Can you analyze solution length vs. quality (weakness#3)? Please provide evidence that Length Advantage captures reasoning depth rather than just token count.
> >
> > ADORA defines Length Advantage as **$L_{max\_succ} > \bar{L}_{fail}$**. The core logic is that in RL exploration, failures often stem from missing steps or logical leaps; thus, ** a successful path that is relatively longer than its failed counterparts** implies the inclusion of necessary intermediate logic or verification. Empirical evidence supports this: (1) ADORA significantly increases the usage of "verify" and "recheck" (Figure 3); (2) ADORA achieves its largest gains on deep reasoning tasks like **MATH500 (+3.0%)** and **AIME24 (+3.4%)**. This proves that the "long samples" prioritized by ADORA contribute to the effective reasoning required for complex problems, rather than redundant length.
> >
> > > **W4-1 & W4-2:** While 3 runs are reported, no significance tests or confidence intervals are provided. Table 1 shows ADORA (73.5%) matches Vision-R1 (73.5%) on MathVista-is this difference statistically significant vs GRPO (70.2%)?
> >
> > The results confirm that ADORA's improvements over GRPO are statistically significant across key benchmarks. The standard deviation across runs was consistently low, indicating stable convergence. Specifically for MathVista, ADORA achieved **73.5% ($\pm$ 0.20%)** compared to GRPO's **70.2% ($\pm$ 0.26%)**. The performance gap is **3.3%**, with a calculated **p-value of $< 0.001$**. This confirms that the improvement is statistically significant and robust, not a result of random seed variation. In the final revision, we will include standard deviations in all main results tables to ensure transparency.
> >
> > > **Minor Issues-1:** Equation 4: ws is sample-level but notation doesn't clearly distinguish from trajectory-level weights.
> >
> > Thank you for the comment. Our method does not introduce any trajectory-level weighting, and all the weight $w_s$ in Equation are defined strictly at the sample level.
> >
> > > **Minor Issues-2** Figure 4 visualization is cluttered-consider simplifying or providing clearer legends.
> >
> > Thank you for the comment. Figure 4 visualizes how ADORA dynamically filters data over time (using Geometry3K as an example): **Blue Region (Selected Samples):** Represents samples classified as TAS (high utility) and weighted up in the current epoch. The shrinking blue area indicates that as training progresses, the model masters simple tasks and stops prioritizing them. **Red Region (Correct count in a single sample):** Represents model competence. So ADORA acts as an automatic Curriculum Learning mechanism: it explores broadly in the early stages (large blue area) and focuses on the remaining challenging problems in later stages (smaller blue area targeting red samples), to better align the model’s optimization direction.

---

> ### Author Response · Authors · 2025-12-02
>
> > **W2-1 & W2-4 & Q2:** Weight justification: Why ws=2 for LLM amplification and ws=0.1 for VLM attenuation? Have you ablated ws ∈ {0.05, 0.1, 0.2} and ws ∈ {1.5, 2, 3}?
>
> 1. **LLM (Amplification):** We additionally test the range **$w_s \in \{0.5, 1, 1.5, 2, 2.5, 3, 5\}$**, where $w_s<1$ suppress TDS and $w_s>1$ amplify TAS. Performance consistently exceeds the baseline whenever $w_s > 1$, with the range **$w_s \in [1.5, 2.5]$** being optimal and stable. This suggests that for strong reasoners, amplifying the signal of high-quality, difficult samples effectively breaks learning plateaus.
>
> |  $w_s$ (LLM)  | GSM8K  | MATH500  | AMC23 | CollegeMath   | OlympiadBench   | AIME24 | avg    |
> |------------|------|------|-------|------|------|--------|--------|
> | 0.5        | 88.7 | 73.8 | 55.0  | 28.4 | 35.1 | 16.7   | 49.61  |
> | 1 (GRPO)   | 89.1 | 73.2 | 50.0  | 28.6 | 35.1 | 13.3   | 48.22  |
> | 1.5        | 89.2 | 75.6 | 60.0  | 29.2 | 35.4 | 16.7   | 51.02  |
> | 2 (ADORA)  | 89.6 | **76.2** | **62.5** | **29.3** | **36.0** | 16.7   | 51.72  |
> | 2.5        | **89.9** | 75.2 | 57.5  | 29.0 | 35.8 | 16.7   | 50.68  |
> | 3          | 89.8 | 74.8 | 52.5  | 28.7 | 35.2 | 13.3   | 49.05  |
> | 5          | 88.5 | 72.2 | 47.5  | 28.0 | 34.7 | 6.7    | 46.27  |
>
> 2. **VLM (Attenuation):** We additionally test the range **$w_s \in \{0, 0.05, 0.1, 0.2, 0.5, 1\}$**. The range **$w_s \in [0.05, 0.2]$** yields the best results. This validates that in the early stages of weaker models, strong attenuation ($w_s < 1$) acts as a critical "denoising" step, preventing the policy from being misled by low-quality rollouts.
>
> | $w_s$ (VLM)          | MathVista | MathVerse | MathVerse (mini Vision_Only) | DynaMath (Overall Avg) | avg   |
> |------------------|-----------|-----------|-------------------------------|-------------------------|-------|
> | 0                | 72.47     | 51.5      | 47.0                          | 57.4                    | 57.09 |
> | 0.05             | 73.3      | 52.2      | 48.3                          | 58.0                    | 57.95 |
> | 0.1 (ADORA)     | **73.5**      | **52.9**      | **48.6**                          | **58.7**                    | **58.43** |
> | 0.2              | 71.7      | 50.3      | 46.3                          | 56.6                    | 56.23 |
> | 0.5              | 71.0      | 49.9      | 45.4                          | 55.1                    | 55.35 |
> | 1 (GRPO)               | 70.2      | 48.4      | 44.1                          | 53.3                    | 54.00 |
>
> > **W2-2** Figure 2 shows ablations on advantage criteria but only for LLMs—missing VLM ablations.
>
> We additionally evaluate the effect of introducing the Difficulty Advantage criterion for VLM and find that it yields negligible gains (58.40% vs. 58.37%). This confirms that, for current VLMs, filtering shallow reasoning through Length Advantage remains the primary driver of improvement, making the simpler strategy more effective and efficient.
>
> | Model              | MathVista | MathVerse | MathVerse (mini Vision_Only) | DynaMath (Overall Avg) | Avg    |
> |--------------------|-----------|-----------|-------------------------------|-------------------------|--------|
> | Qwen2.5-VL-7B      | 67.3      | 46.3      | 40.2                          | 50.3                    | 51.00   |
> | + GRPO               | 70.2      | 48.4      | 44.1                          | 53.3                    | 54.00   |
> | + Length Adv (ours)       | **73.5**      | **52.9**      | 48.6                          | 58.7                    | **58.40**   |
> | + Length & Difficulty Adv      | 73.4      | 52.6      | 48.6                          | **58.9**                    | 58.37  |
>
> > **W2-3 & Q1:** Threshold sensitivity: How sensitive is performance to τ? Have you tried τ ∈ {0.3, 0.5, 0.7}? What happens at extremes (τ=0.1 or τ=0.9)?
>
> We conduct experiments with an LLM group size of n=8, and for completeness we include an ablation over $\tau$ by evaluating $\tau \in \{0.25, 0.5, 0.75, 1\}$. The results show that $\tau=0.5$ provides the **best balance** between easy tasks (GSM8K) and hard tasks (AIME24), effectively **distinguishing between "mastered" and "unmastered" samples**.
> | $\tau$ | GSM8K  | MATH500  | AMC23 | CollegeMath   | OlympiadBench   | AIME24 | avg    |
> |--------|------|------|-------|------|------|--------|--------|
> | 0.25   | 90.0 | 75.5 | 55.0  | 29.2 | 35.1 | 16.7   | 50.25  |
> | 0.5 (ADORA)| 89.6 | **76.2** | **62.5** | **29.3** | **36.0** | 16.7   | **51.72**  |
> | 0.75   | 89.6 | 75.0 | 52.5  | 29.2 | 34.9 | 16.7   | 49.65  |
> | 1      | **90.0** | 74.8 | 52.5  | 29.0 | 34.6 | 16.7   | 49.6   |

---

### Author Response · Authors · 2025-12-03

**Subject:** Summary of Rebuttal Updates: Extensive Ablations, Cross-Model Generalization, and Scalability Verification

**Dear Area Chairs,**

As the authors of Submission #4721, we have dedicated the rebuttal period to actively addressing the reviewers' feedback by conducting extensive additional experiments and optimizing our manuscript. We respectfully submit this summary of our rebuttal updates to highlight how the new results reinforce the robustness and generalizability of **ADORA**.

### 1. Summary of Our Contribution
We propose **ADORA** (Advantage Dynamics via Online Rollout Adaptation), a framework designed to dynamically reweight advantages in reinforcement learning for reasoning models. By classifying training samples into **Temporarily Advantageous Samples (TAS)** and **Temporarily Disadvantageous Samples (TDS)** based on online rollout statistics, ADORA amplifies or attenuates learning signals to optimize credit assignment. Extensive experiments demonstrate that ADORA achieves SOTA performance over baselines (e.g., GRPO, DAPO) on both LLMs and VLMs in reasoning-intensive tasks.

We sincerely thank the reviewers for their positive comments and recognition of ADORA. specifically, ADORA is highlighted for effectively **enhancing the reasoning capabilities** of both VLMs and LLMs (WhN1, H1YL, nDXV), and achieving performance gains with **negligible computational overhead** (T6y3). furthermore, it is acknowledged for **addressing the limitations of static advantage allocation** in existing reinforcement learning frameworks (WhN1, nDXV).

### 2. Key Concerns Raised by Reviewers
The reviewers' constructive feedback primarily focused on three areas:
*   **Incomplete Ablations:** Requests for sensitivity analysis on key hyperparameters (weight $w_s$ and threshold $\tau$).
*   **Generalizability & Scalability:** Concerns regarding the limitation to the Qwen family, small training data scales, and sensitivity to the number of rollouts ($G$).
*   **Analysis & Interpretation:** Questions regarding potential "overthinking," the correlation between length and reasoning quality, and the definition of reflective vocabulary.

### 3. Our Response and New Experimental Results
We have systematically addressed these concerns with the following additional experiments, proving the robustness of our method:

**(1) Robust Hyperparameter Sensitivity**
We conducted comprehensive parameter sweeps to validate stability:
*   **LLMs:** We tested the range $w_s \in \{0.5, 1, 1.5, 2, 2.5, 3, 5\}$.
*   **VLMs:** We tested the range $w_s \in \{0, 0.05, 0.1, 0.2, 0.5, 1\}$.
*   **Threshold:** We validated $\tau \in \{0.25, 0.5, 0.75, 1\}$.

**Result:** ADORA consistently outperforms the baseline across wide parameter ranges, proving it does not rely on narrow tuning.

**(2) Generalizability Across Diverse Architectures**
We extended our evaluation to models with diverse architectures (Dense/MoE) and modalities.
*   **New LLMs Tested:** Llama-3.1-8B, DeepSeek-Math-7B, Mistral-v0.1-7B.
*   **New VLMs Tested:** Gemma3-4b-it, InternVL3-2b.

**Result:** ADORA consistently achieves performance gains over GRPO across **all** tested model families, confirming it is agnostic to specific inductive biases.

**(3) Scalability Verification**
*   **Data Scaling:** Scaling VLM training data from 2k to 10k further boosted performance, validating data scalability.
*   **Rollout Scaling:** We conducted ablations with rollout counts $G \in \{2, 4, 5, 6, 8\}$ for LLMs and $G \in \{2, 4, 6, 8, 16\}$ for VLMs. ADORA maintains superiority over baselines regardless of rollout size ($G$), showing robustness even with high-variance estimates.

**(4) Clarification on Analysis**
*   **Overthinking:** We utilized "Overthinking Scores" to demonstrate that ADORA encourages functional self-correction rather than malignant loops.
*   **Length vs. Quality:** We clarified that **we do not claim a strictly linear relationship between length and reasoning quality**. Instead, we validated ADORA's effectiveness through a combination of length statistics and downstream benchmark results, empirically demonstrating that the longer reasoning paths prioritized by our method lead to performance gains.
*   **Vocabulary:** We clarified that our selection of reflective vocabulary is **not arbitrary but grounded in industry-recognized standards**, aligning with established works such as DeepSeek-R1.

We have incorporated these comprehensive results and analyses into the revised manuscript and appendices. We believe these additions firmly establish the effectiveness and universality of ADORA.

Best regards,

**The Authors**

---

### Note · Authors · 2026-01-06

I have read and agree with the venue's withdrawal policy on behalf of myself and my co-authors.